# On Evaluating Policies for Robust POMDPs

**Merlijn Krale**[*]
Radboud University
Nijmegen, The Netherlands
merlijn.krale@ru.nl

**Eline M. Bovy**[*]
Radboud University
Nijmegen, The Netherlands
eline.bovy@ru.nl

**Maris F. L. Galesloot**[*]
Radboud University
Nijmegen, The Netherlands
maris.galesloot@ru.nl

**Thiago D. Simão**
Eindhoven University of Technology
Eindhoven, The Netherlands
t.simao@tue.nl

**Nils Jansen**
Ruhr-University Bochum & Radboud University
Bochum, Germany & Nijmegen, The Netherlands
n.jansen@rub.de

## Abstract

Robust partially observable Markov decision processes (RPOMDPs) model sequential decision-making problems under partial observability, where an agent must be *robust* against a range of dynamics. RPOMDPs can be viewed as a two-player game between an agent, who selects actions, and *nature*, who adversarially selects the dynamics. Evaluating an agent policy requires finding an adversarial nature policy, which is computationally challenging. In this paper, we advance the evaluation of agent policies for RPOMDPs in three ways. First, we discuss suitable benchmarks. We observe that for some RPOMDPs, an optimal agent policy can be found by considering only subsets of nature policies, making them easier to solve. We formalize this concept of *solvability* and construct three benchmarks that are only solvable for expressive sets of nature policies. Second, we describe a new method to evaluate agent policies for RPOMDPs by solving an equivalent MDP. Third, we lift two well-known upper bounds from POMDPs to RPOMDPs, which can be used to efficiently approximate the optimality gap of a policy and serve as baselines. Our experimental evaluation shows that (1) our proposed benchmarks cannot be solved by assuming naive nature policies, (2) our method of evaluating policies is accurate, and (3) the upper bounds provide solid baselines for evaluation.

## 1 Introduction

Partially observable Markov decision processes [POMDPs; 25] are ubiquitous for representing sequential decision-making problems under partial observability. POMDPs are used to represent real-world problems, such as robotics [31], infrastructure maintenance [39, 38], and wildlife conservation [13]. Yet, to model such problems, the dynamics of the problem need to be precisely known, which is often unrealistic [26, 57]. Robust MDPs [RMDP; 23, 43, 55] capture *model uncertainty* as a two-player game between the agent, who picks actions, and nature, who adversarially picks the dynamics of the model. RMDPs can effectively represent model uncertainty arising in reinforcement learning [54, 11, 37] or from abstraction [1, 2, 32], but do not account for partial observability. Robust POMDPs [RPOMDPs; 45] extend RMDPs with partial observability, and are thus more expressive. However, finding and evaluating policies for RPOMDPs is hard: existing solvers must use an extensive range of approximations to find and evaluate policies within reasonable time [53, 15, 9].

To test such RPOMDP solvers, we require an *evaluation pipeline*, such as shown in Figure 1. To start, we need a set of *benchmarks* that allows us to investigate the limits of the solver. Once the solver

---

[*]Joint main authorship.

39th Conference on Neural Information Processing Systems (NeurIPS 2025).

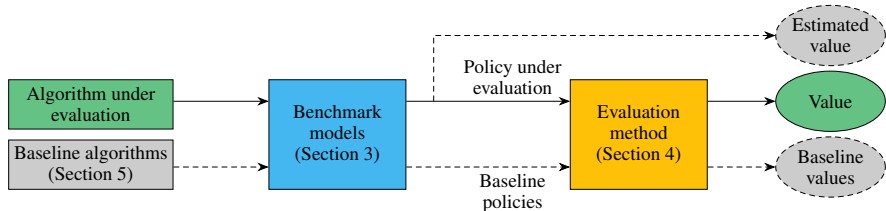

Figure 1: Visualisation of the evaluation pipeline: benchmarks, an evaluation method, and baselines.

computes a policy for such benchmarks, we need an *evaluation method* to obtain a value for the policy. Finally, we compare this value with that of suitable *baselines*, and, if available, the estimated value of the policy as provided by the solver. In summary, an evaluation pipeline requires at least these three parts: benchmarks, an evaluation method, and baselines.

Most existing RPOMDP literature does not address all these parts. Firstly, the benchmarks used are often existing POMDPs extended with artificial model uncertainty [45, 9, 15, 42, 22], and it is unclear if such benchmarks are representative of RPOMDPs in general. Secondly, policy evaluation requires finding a worst-case nature policy, which can be prohibitively expensive. Existing work often either simplifies the evaluation process or reports the estimated value of the algorithm without explicitly evaluating the policy. Lastly, there are no readily computable baselines for RPOMDPs, hindering the efficient assessment of new solvers. In this paper, we aim to address all three parts of the evaluation pipeline with the following contributions:

**(1) We propose novel solvability classes and corresponding benchmark problems (Section 3).** We discuss the problem of defining benchmarks, and argue that suitable benchmarks should test whether a solver has considered all possible nature policies. Thus, if we can find an optimal agent policy for an RPOMDP by considering only naive subsets of nature policies, then the RPOMDP is not a suitable benchmark. We formalize this intuition using the concept of *solvability*. Based on this concept, we define three small benchmarks for which we prove that the optimal agent policy cannot be found against naive subsets of nature policies.

**(2) We propose a new robust policy evaluation method (Section 4).** To evaluate agent policies, we propose to compute a *best-response nature policy*, i.e., the worst-case nature policy against the policy under evaluation. We show that this evaluation method is less computationally expensive than finding an optimal nature policy, and we prove that the aforementioned best-response policy is, in fact, the worst-case evaluation for a fixed agent policy. Lastly, we demonstrate that this evaluation method can be represented in general as an MDP with continuous state and action spaces, enabling evaluation with off-the-shelf (approximate) continuous MDP solvers.

**(3) We define two new efficient robust baselines (Section 5).** We lift two existing POMDP approximation algorithms, QMDP [33] and the fast informed bound (FIB) [21], to RPOMDPs, which we denote as *robust QMDP* (RQMDP) and *robust FIB* (RFIB), respectively. We prove the convergence and relative tightness of these approximations and highlight that, under conventional technical assumptions, both are tractable baselines to compute.

Finally, we conduct an empirical evaluation that (1) shows that the new benchmarks cannot be solved by assuming naive nature policies, (2) validates the accuracy of our evaluation method, and (3) demonstrates the applicability of our approximations as baselines. We implement all methods in a Julia framework (based on `POMDPs.jl` [12]) to facilitate future research; available on Zenodo [29].

## 2   Preliminaries: Robust POMDPs

This section gives a comprehensive definition of RPOMDPs. We first introduce some basic notation. We denote the set of all possible probability distributions over the (countable) set $A$ as $\Delta(A)$, and all elements with non-zero probability in a distribution $\mu \in \Delta(A)$ as $\mathrm{supp}(\mu)$. For any predicate $P$, the *Iverson brackets* $[P]$ return 1 if $P$ is true and 0 otherwise. Given a convex set $C$, we denote the set of all extreme points as $\mathrm{Extremes}(C)$, and the centroid of the set, i.e., the arithmetic mean of all points in $C$, as $\mathrm{Centroid}(C)$. Lastly, given a function $f \colon X \to \Delta(Y)$ and elements $x \in X$, $y \in Y$,

$f(y \mid x)$ denotes the probability of element $y$ according to $f(x)$, and $y \sim f(x)$ denotes a randomly sampled element $y$ from $f(x)$.

Different definitions of RPOMDPs exist, as discussed in [3]. We focus on a variant of their definition, with some additional assumptions for simplicity.

**Definition 1** (RPOMDP). *An RPOMDP is a tuple* $\mathcal{M} = \langle S, A, \Omega, \mathcal{U}, \mathcal{T}, \mathcal{O}, R, b_0, \gamma \rangle$, *with:*

- $S$, $A$ *and* $\Omega$ *(finite) sets of* states, actions *and* observations;
- $\mathcal{U} \subseteq \{f \colon \mathrm{Var} \to \mathbb{R}\} = \mathbb{R}^{|\mathrm{Var}|}$ *the* uncertainty set, *with* Var *a finite set of* decision variables;
- $\mathcal{T} \colon \mathcal{U} \to (S \times A \to \Delta(S))$ *the* uncertain transition function, *which defines a* transition function $T \colon S \times A \to \Delta(S)$ *for each assignment of decision variables* $u \in \mathcal{U}$;
- $\mathcal{O} \colon \mathcal{U} \to (S \times A \times S \to \Delta(\Omega))$ *the* uncertain observation function *which defines an* observation function $O \colon S \times A \times S \to \Delta(\Omega)$ *for each assignment* $u \in \mathcal{U}$;
- $R \colon S \times A \to \mathbb{R}$ *the* reward function;[1]
- $b_0 \in \Delta(S)$ *the* initial state distribution *(or* initial belief*);*
- $\gamma \in [0, 1)$ *the* discount factor.

For notational convenience, we define the joint uncertain transition-observation function $\mathcal{P} \colon \mathcal{U} \to (S \times A \to \Delta(S \times \Omega))$, with $\mathcal{P}(u)(s', o \mid s, a) = \mathcal{T}(u)(s' \mid s, a)\mathcal{O}(u)(o \mid s, a, s')$. Note that an RPOMDP with a singleton uncertainty set is a POMDP, a fully observable RPOMDP is a robust MDP [RMDP; 23, 43, 55], and an RMDP with a singleton uncertainty set is an MDP [48].

Intuitively, RPOMDPs describe a zero-sum game between an agent and nature, with policies $\pi$ and $\theta$, respectively [3]. The game starts in a state $s_0 \in S$ as sampled from the initial state distribution $b_0$. At each timestep $t \in \mathbb{N}$, the agent picks an action $a_t \in A$ according to its policy $\pi$. Then, nature picks a decision variable assignment $u_t \in \mathcal{U}$ according to policy $\theta$. Next, the environment transitions to a state $s_{t+1} \sim \mathcal{T}(u_t)(s_t, a_t)$, and emits an observation $o_t \sim \mathcal{O}(u_t)(s_t, a_t, s_{t+1})$. Lastly, the agent and nature receive a reward $r_t = R(s_t, a_t)$, which is not observed.

**Policies.** Next, we introduce notation for memory-based randomized agent policies. Let $X \subset \mathbb{R}^N$, with $N \in \mathbb{N}$, denote a *memory space*. Possible memory spaces include the set of beliefs (i.e. $X \subseteq \Delta(S)$, as used in [45]) or the set of memory nodes of a finite state controller (as used in [9, 15]. Then, we define the set of agent policies as $\Pi = \{\langle \sigma \colon X \to \Delta(A), \tau \colon X \times A \times \Omega \to \Delta(X) \rangle\}$, where $\sigma$ denotes the *action selection function*, and $\tau$ the *memory update function*. Given a policy $\pi = \langle \sigma, \tau \rangle \in \Pi$ and a *memory state* $x_t \in X$, the agent chooses action $a_t \sim \sigma(x)$, and updates it's internal memory state to $x_{t+1} \sim \tau(x_t, a_t, o_{t+1})$.

The formalization of our nature policies is based on two assumptions. First, we assume *dynamic uncertainty* [23] (or *zero stickiness* [3]), which means nature can choose a different decision variable assignment when revisiting a state-action pair. In contrast, *static uncertainty* (or *full stickiness*) refers to the setting where nature must always choose the same decision variable assignment. Second, we assume nature has knowledge of the history of (1) the visited states, (2) the agent's memory states, and (3) the agent's actions.[2] Combining these assumptions, the set of nature policies is given as $\Theta = \{\theta \colon (S \times X \times A)^N \to \mathcal{U} \mid N \in \mathbb{N}\}$. Note that weakening either assumption will yield a more restrictive set of nature policies; thus, these assumptions provide a lower bound for any such setting.

**Structural Assumptions.** We make two additional structural assumptions. First, we assume the uncertainty set is $\langle s, a \rangle$-*rectangular*. Intuitively, nature can choose probabilities for any state-action pair independently. This means we assume that the set of decision variables *Var* can be partitioned into smaller sets $Var_{s,a}$, such that (1) the uncertainty set is $\mathcal{U} = \times_{s,a \in S \times A} \{Var_{s,a} \to \mathbb{R}\}$, and (2) any decision variable in $Var_{s,a}$ only influences state-action pair $\langle s, a \rangle$. This assumption is common in the RPOMDP literature and, similar to the assumptions above, provides a lower bound for any non-rectangular setting. Lastly, we assume the set of joint transition-observation functions in $\mathcal{P}$ given the uncertainty set $\mathcal{U}$, i.e., $\{\mathcal{P}(u) \mid u \in \mathcal{U}\}$, is convex, closed, and graph-preserving. Convexity is a common assumption for tractability reasons, while closedness and graph-preservation are sufficient to guarantee the existence of an optimal nature policy[3] [36], which we define below.

---

[1]This definition could trivially be extended to include uncertainty in the reward function [45].

[2]Nature does not require knowledge of observations, since these only affect the agent's memory states.

[3]An RPOMDP given an agent policy induces an infinite action POMDP for nature. Since nature has full observability, this can alternatively be interpreted as an RMDP with only a single agent action. Thus, Theorem 2 from Meggendorfer et al. [36] applies, and we know optimal agent and nature policies exist.

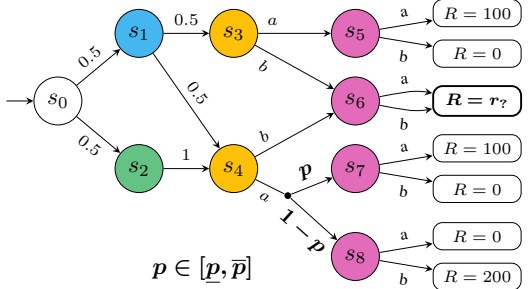

| Solvability | $\underline{p}$ | $\overline{p}$ | $r_?$ |
|---|---|---|---|
| Trivially | 0.2 | 0.6 | 80 |
| Center | 0.1 | 0.6 | 80 |
| Entropy | 0.1 | 0.6 | 80 |
| RMDP | 0.1 | 0.6 | 80 |
| Stationary | 0.1 | 0.9 | 80 |
| None of the above | 0.1 | 0.9 | 70 |

Figure 2: Example RPOMDP with parameter values for solvability classes. We use TOY$^*$ to denote the variant of this RPOMDP with the parameter values from "None of the above".

**Objective.** The agent's objective is to maximize its *value*, i.e., the infinite-horizon expected cumulative discounted reward. We assume nature is *adversarial*, meaning it aims to minimize the value. Let $V^{\pi,\theta} := \mathbb{E}_{\pi,\theta}\left[\sum_{t=0}^{\infty} \gamma^t r_t \mid s_0 \sim b_0\right]$ denote the value of the RPOMDP given policies $\pi \in \Pi$ and $\theta \in \Theta$. We denote the optimal value for the agent and nature as $V^{\mathfrak{a}} = \max_{\pi \in \Pi} \min_{\theta \in \Theta} V^{\pi,\theta}$ and $V^{\mathfrak{n}} = \min_{\theta \in \Theta} \max_{\pi \in \Pi} V^{\pi,\theta}$, respectively, with corresponding optimal agent- and nature policies $\pi^*$ and $\theta^*$. If $V^{\mathfrak{a}} = V^{\mathfrak{n}}$, then the underlying game has a *Nash equilibrium* [46], in which case $\pi^*$ and $\theta^*$ are called *Nash policies*. In this paper, we focus on the agent's objective $V^{\mathfrak{a}}$, that is, finding and evaluating agent policies with respect to their *worst-case* nature policies. We note that this is related to the notion of *Stackelberg equilibria* [46]. As for POMDPs [35], solving infinite-horizon discounted RPOMDPs is undecidable. Thus, in general, agent policies can be $\varepsilon$-optimal at best.

## 3 Solvability Classes and Benchmark Models

In this section, we first propose the concept of *solvability* to analyze the complexity of an RPOMDP. Next, we introduce three small RPOMDP benchmarks that are complex according to our definition. In Section 6, we provide an empirical evaluation of the complexity of these benchmarks, which we compare to standard POMDPs extended with artificial uncertainty sets.

We introduce the TOY environment (Figure 2, left). Throughout this section, we will use variants of this model with different choices for the reward $r_?$ and uncertainty set $[\underline{p}, \overline{p}]$ (Figure 2, right). For notational convenience, we assume $\gamma = 1$. Intuitively, the agent needs to make two decisions. Firstly, in the 🟡 states, the agent must choose between the safe action $b$, which will always yield a value $r_?$, and the more risky action $a$. If $a$ is selected, the agent must make another choice between $a$ and $b$ in the 🟣 states. Which action is optimal in the 🟡 and 🟣 states depends both on the history up to that point, as this indicates in which states the agent can be, as well as the nature policy.

We highlight two different agent policies for this environment. The *safe* policy $\pi_s^*$ picks the safe action $b$ in the 🟡 states regardless of the previous observation, which yields a guaranteed value of $r_?$. In contrast, the *risky* policy $\pi_r^*$ picks the riskier action $a$ in the 🟡 states if the previous observation was 🟢, and the safe action $b$ otherwise. Furthermore, $\pi_r^*$ picks action $b$ in the 🟣 states. For $r_? = 80$, $\pi_s^*$ is optimal if $\underline{p} \le 0.6 \le \overline{p}$, and $\pi_r^*$ is optimal if $\underline{p} \le 0.6$ and $0.2 \le \overline{p} \le 0.6$. Similarly, for $r_? = 70$, $\pi_s^*$ is optimal if $\underline{p} \le 0.4$ and $0.65 \le \overline{p}$. We provide a proof in Appendix A.1.

### 3.1 Solvability and $\Theta$-solvable

We define the concept of *solvability* as follows:

**Definition 2.** *Recall that $\Theta$ and $\Pi$ denote the sets of all nature- and agent policies. Define $\Pi^{\overline{\Theta}}$ as the set of agent policies that are optimal against both $\Theta$, as well as a subset of nature policies $\overline{\Theta} \subseteq \Theta$:*

$$\Pi^{\overline{\Theta}} = \underset{\pi \in \Pi}{\mathrm{argmax}} \min_{\overline{\theta} \in \overline{\Theta}} V^{\pi,\overline{\theta}} \cap \underset{\pi \in \Pi}{\mathrm{argmax}} \min_{\theta \in \Theta} V^{\pi,\theta}. \tag{1}$$

*Then, a model $\mathcal{M}$ is $\overline{\Theta}$-solvable if $\Pi^{\overline{\Theta}} \neq \emptyset$.*

Intuitively, a model is $\overline{\Theta}$-solvable if there exists at least one optimal agent policy against the subset of nature policies $\overline{\Theta}$ that is also optimal against all nature policies $\Theta$. For such models, solvers that are

only robust against $\overline{\Theta}$ could still find optimal agent policies. For some subsets $\overline{\Theta}$, this property may be undesirable for benchmarks, as we will discuss below. Note that $\overline{\Theta}$-solvability does not imply that $\theta^* \in \overline{\Theta}$, nor that the value against these sets is equal. Below, we define different *solvability classes* based on whether a model is solvable for a particular $\overline{\Theta}$.

**Trivial solvability.** We first consider the most extreme case. A model is *trivially solvable* if it is solvable for *any* set $\overline{\Theta}$. Such models are unsuitable as benchmarks, since they do not adequately test whether a solver has considered all possible nature policies.

**Example 1.** TOY *with uncertainty set $p \in [0.2, 0.6]$ and reward $r_? = 80$ is trivially solvable, since $\pi_r^*$ is optimal for any choice of $\theta$.*

**Naive solvability.** Next, we consider models where the optimal value does depend on the choice of the nature policy $\theta$, but where a naive choice of $\theta$ suffices. Such models are not adequate to show the capabilities of solvers to be robust against all possible nature policies, and are thus unsuitable as benchmarks. We define the following three nature policies that induce a naive solvability class:

- $\theta_{\text{Center}}$, which picks decision variables such that $\mathcal{P}(u) = \text{Centroid}\left(\{\mathcal{P}(u) \mid u \in \mathcal{U}\}\right)$. For RPOMDPs constructed by extending a POMDP with model uncertainty, this nature policy often corresponds to the original POMDP.
- $\theta_{\text{Ent}}$, which picks decision variables to maximize the *entropy* of the probability distribution of each transition. This nature policy generally results in high uncertainty for the agent.
- $\theta_{\text{RMDP}}$, which picks decision variables that are optimal in the underlying RMDP, i. e., optimal against a fully observing agent policy. In particular, this nature policy is good if partial observability has little effect on the optimal agent policy.

Thus, we call a model *naively solvable* if it is either $\{\theta_{\text{Center}}\}$-, $\{\theta_{\text{Ent}}\}$-, or $\{\theta_{\text{RMDP}}\}$-solvable. This list is not exhaustive; additional solvability classes can be defined as naive if so desired.

**Example 2.** TOY *with $p \in [0.1, 0.6]$ and $r_? = 80$ is not trivially solvable, since $\pi_r^*$ is optimal against $\Theta$, but is not optimal for each individual $\theta$. For example, if $\theta(\cdot) = \{p \mapsto 0.1\}$, the agent can gain a higher value by choosing the risky action $a$ in the 🟡 states and action $b$ in the 🟣 states, regardless of the observation history. However, $\pi_r^*$ is optimal against all the naive policies $\theta_{Center}(\cdot) = \{p \mapsto 0.35\}$, $\theta_{Ent}(\cdot) = \{p \mapsto 0.5\}$, and $\theta_{RMDP}(\cdot) = \{p \mapsto 0.6\}$, which means the model is naively solvable.*

**Stationary solvability.** Lastly, a model is *stationary solvable* if it is solvable for the set of stationary nature policies $\Theta^{\text{Sta}}$. A stationary nature policy chooses the same variable assignment for all histories, and can therefore be respresented by a single variable assignment: $\Theta^{\text{Sta}} = \mathcal{U}$. Intuitively, an RPOMDP with a stationary nature policy induces a POMDP with the same state space as the RPOMDP. In contrast to the previous solvability classes, the set of nature policies remains continuous. We are not aware of a way to exploit the stationary solvability of a model to solve it. Thus, we do not consider stationary solvable models unsuitable benchmarks.

**Example 3.** *Consider* TOY *with $p \in [0.1, 0.9]$ and $r_? = 80$. Using similar logic as above, against all naive nature policies $\theta \in \{\theta_{Center}, \theta_{Ent}, \theta_{RMDP}\}$, it is optimal for the agent to take the risky action $a$ in the 🟡 states for one of the possible observation histories. However, for any agent policy that takes a risky action in 🟡, there exists a stationary nature policy that yields a lower value than $r_?$. Thus, $\pi_s^*$ is the only optimal policy against both $\Theta^{Sta}$ and $\Theta$, meaning the model is stationary solvable and not naively solvable. In contrast, if we use the same uncertainty set with $r_? = 70$, taking the risky action $a$ in 🟡 is optimal against all stationary nature policies, while $\pi_s^*$ is optimal against the set of non-stationary policies. Therefore, the model is not stationary solvable.*

## 3.2 Benchmarks

We introduce three novel benchmark environments that, according to our classification above, require solving against expressive nature policies. The first, denoted TOY$^*$, is the variant of the TOY RPOMDP (Figure 2) that does not fall within any of our solvability classes, i.e., with parameter values from "None of the above". In the following, we provide two more benchmarks.

**ECHO.** The ECHO environment (Figure 3, left) is inspired by predictive maintenance problems. Starting in one of two distinguishable states $x$ or $y$, the agent picks an action $a_i$ with $i \in \{x, y\}$. Based on this action, the agent transitions to the state $n_i$, and can then take the *echo (e)* action to transition to state $i$. However, the machine has a probability $\delta$ to transition to the broken states $x'$ or

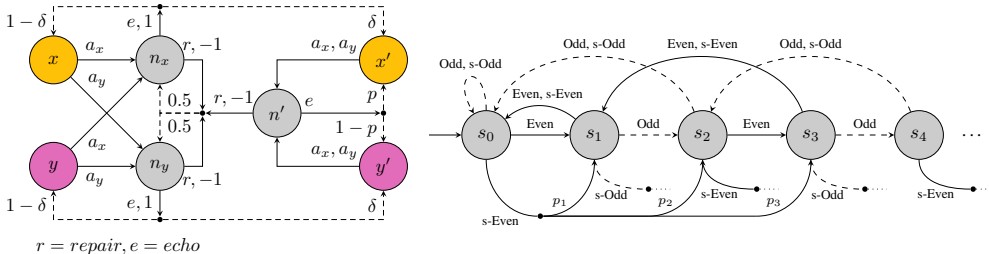

Figure 3: Visualizations of the ECHO environment (left) and PARITY environment (right).

$y'$ instead, which return the same observations as $x$ and $y$. From these, the agent always transitions to state $n'$, where the outcome of the echo action is determined by a parameter $p \in [\underline{p}, \bar{p}]$. The agent receives a reward of 1 for taking the echo action in $n_x$ or $n_y$, but none in $n'$. Alternatively, they may take a *repair (r)* action in any of these states at a cost of $-1$, which returns the agent to $n_x$ or $n_y$ with probability 0.5. We make the following claim about this environment:

**Theorem 1.** ECHO, *with* $\underline{p} = 0.01$, $\bar{p} = 0.99$, $\delta = 0.1$ *and* $\gamma = 0.95$, *is not in any of the solvability classes defined in Section 3.1.*

We give a more general formulation of this theorem, as well as a proof, in Appendix A.2.1. Intuitively, a non-stationary nature policy can pick $p$ depending on whether the agent previously played $a_x$ or $a_y$, while a stationary nature cannot. Thus, the optimal policy will sometimes repair even if it is uncertain whether the machine is broken, while this is always suboptimal against a stationary nature policy.

**PARITY.** Lastly, we consider an abstract chain environment called PARITY($N$), parameterized with $N \in \mathbb{N}_{>0}$, as visualized in Figure 3 (right). The environment consists of a chain of indistinguishable states $s_0$ to $s_N$, and the agent receives a reward when reaching $s_N$. To do so, the agent can guess the *parity* of its current state $s_i$, and can choose actions with either *deterministic (Even, Odd)* or *stochastic (s-Even, s-Odd)* outcome. When correctly guessing the parity in state $s_i$, the agent moves to state $s_{i+1}$ when choosing deterministic actions, and to states $s_{i+1}, s_{i+2}$ or $s_{i+3}$ with probabilities $p_1 \in P_1, p_2 \in P_2, p_3 \in P_3$ otherwise. However, the agent moves to state $s_{i-2}$ for incorrect guesses. Stochastic actions take the agent further on average, but make it harder to guess correctly in the future. In addition to finite-length chains, we consider a chain of infinite length with the same dynamics, denoted PARITY($\infty$), where the agent receives an immediate reward equal to the number of steps they take. We show this problem can be reduced to a 9-state model in Appendix A.2.2. For this environment, we show the following with regard to solvability:

**Theorem 2.** PARITY($\infty$), *with* $P_1 = \{0.2\}$, $P_2 = [0.1, 0.7]$, $P_3 = [0.1, 0.7]$, *and* $\gamma \geq 0.7\bar{3}$, *is not naively solvable.*

We provide a proof in Appendix A.2.2. Intuitively, our proof shows that optimal policies for naive subsets of nature policies take the riskier stochastic actions, which is suboptimal in the worst case. We empirically show in Section 6 that PARITY(10), with with $P_1 = \{0.1\}$, $P_2 = [0.5, 0.8]$, $P_3 = [0.1, 0.4]$, and $\gamma = 0.95$ is also not naively solvable.

## 4 Evaluating Agent Policies in RPOMDPs

In this section, we consider the problem of policy evaluation for RPOMDPs. To evaluate an agent policy $\pi = \langle \sigma, \tau \rangle \in \Pi$, we must consider against which nature policy to evaluate. One choice is, if it exists, to evaluate against the optimal nature policy $\theta^*$, the worst-case nature policy when considering all possible agent policies. However, reasoning over all possible agent policies is often intractable, and is unnecessary if we will only use $\theta^*$ to evaluate a single policy. Moreover, there might be alternative nature policies that yield lower values, as the following example illustrates.

**Example 4.** *Consider an agent policy in the* TOY$^*$ *environment (Figure 2) that always picks action $a$ in the 🟡 states, then $b$ in the 🟣 states. This policy achieves a close-to-optimal value of $66\frac{2}{3}$ against $\theta^*$, but is highly exploitable by a nature policy that chooses $p = 0$, achieving a value of $0$.*

Since evaluation against $\theta^*$ does not always give a good indication of the robustness of a policy, we propose to evaluate the policy $\pi$ against its own worst-case nature, which we denote as the *best-response* policy $\theta_\pi^* \in \mathrm{argmin}_{\theta \in \Theta} V^{\pi,\theta}$. In practice, finding (approximations of) $\theta_\pi^*$ is computationally cheaper than finding $\theta^*$. In particular, we show that for $\langle s, a \rangle$-rectangular and dynamic uncertainty models, the policy type of $\theta_\pi^*$ is typically simpler than that of $\theta^*$, which suggests that it is easier to find. Recalling $\mathcal{P}$ and Extremes $(\cdot)$ from the preliminaries, we state this as follows:

**Theorem 3.** *For any agent policy $\pi \in \Pi$ there exists a best-response nature policy $\theta_\pi^* \colon X \to \mathcal{U}$ such that $\forall x \in X \colon \mathcal{P}\big(\theta_\pi^*(x)\big) \in$ Extremes $(\{\mathcal{P}(u) \mid u \in \mathcal{U}\})$.*

We provide a full proof in Appendix B. Intuitively, nature does not need to consider the whole history since, due to $\langle s, a \rangle$-rectangularity, it can pick variable assignments that are optimal for any visited state-action pair. However, nature's decision should still be based on what the agent will do in the future, which depends on the agent's current memory state $x \in X$, as the agent policy is stationary over $X$ and the memory update $\tau$ is fixed for a policy $\pi = \langle \sigma, \tau \rangle \in \Pi$. Moreover, since the agent policy is fixed, nature enjoys more expressive power in the variable assignments it chooses. Therefore, nature only needs to consider a subset of the policy class $\Theta$ to select the best-response actions.

Now, using Theorem 3, we propose a method for finding $\theta_\pi^*$. In particular, since $\theta_\pi^*$ is Markovian in $X$, we can represent our problem as a (possibly continuous-space) *nature MDP* with state space $S_\mathfrak{n} = X$, action space $A_\mathfrak{n} \subset \mathcal{U}$, and dynamics that represent both the underlying RPOMDP and the memory update of the agent. To simplify our notation, we use an expanded state space that explicitly includes the current state and agent action. In that case, we define the nature MDP as follows:

**Definition 3** (Nature MDP). *Assume we have an RPOMDP $\mathcal{M} = \langle S, A, \Omega, \mathcal{U}, \mathcal{T}, \mathcal{O}, R, b_0, \gamma \rangle$ and an agent policy tuple $\pi = \langle \sigma, \tau \rangle \in \Pi$ that uses a (possibly continuous) memory space $X$. Then, the corresponding nature MDP $M_\mathfrak{n}^\pi$ is defined as $M_\mathfrak{n}^\pi = \langle S_\mathfrak{n}, A_\mathfrak{n}, T_\mathfrak{n}, R_\mathfrak{n}, \mu_\mathfrak{n}, \gamma \rangle$, with:*

- *$S_\mathfrak{n} = S \times X \times A$ the state space;*
- *$A_\mathfrak{n} = \{u \in \mathcal{U} \mid \mathcal{P}(u) \in$ Extremes $(\{\mathcal{P}(u) \mid u \in \mathcal{U}\})\}$ the action space.*
- *$T_\mathfrak{n} \colon S_\mathfrak{n} \times A_\mathfrak{n} \to \Delta(S_\mathfrak{n})$ the transition function, defined as:*

$$T_\mathfrak{n}(\langle s', x', a' \rangle \mid \langle s, x, a \rangle, a_\mathfrak{n}) = \sigma(a' \mid x') \sum_{o \in \Omega} \mathcal{P}(a_\mathfrak{n})(s', o \mid s, a)\tau(x' \mid x, a, o).$$

- *$R_\mathfrak{n} \colon S_\mathfrak{n} \to \mathbb{R}$ the state-based reward function, with $R_\mathfrak{n}(\langle s, x, a \rangle) = R(s, a)$.*
- *$\mu_\mathfrak{n} \in \Delta(S_\mathfrak{n})$ the initial state distribution, with $\mu_\mathfrak{n}(\langle s, x, a \rangle) = \sigma(a \mid x)[x = x_0]b_0(s)$, where $x_0 \in X$ is the initial memory for policy $\pi$, i.e. $x_0 = b_0$ if the policy is belief-based.*

**Remark 1.** *For any nature state $s_\mathfrak{n} = \langle s, x, a \rangle \in S_\mathfrak{n}$, transitions only depend on decision variables in the set $\mathrm{Var}_{s,a}$. Thus, in practice, the action space $A_\mathfrak{n}$ in the nature MDP $M_\mathfrak{n}^\pi$ can be represented using state-dependent action sets $A_\mathfrak{n}(s_\mathfrak{n})$ of much smaller size, i.e., $|A_\mathfrak{n}(s_\mathfrak{n})| \ll |A_\mathfrak{n}|$.*

The action space $A_\mathfrak{n}$ of the nature MDP $M_\mathfrak{n}^\pi$ is a subset of the action space of nature in the corresponding RPOMDP $\mathcal{M}$. Therefore, the set of policies for $M_\mathfrak{n}^\pi$, denoted $\Theta_\mathfrak{n}^\pi := \{\theta \colon S_\mathfrak{n} \to A_\mathfrak{n}\}$, are also valid nature policies for $\mathcal{M}$. Let $V_\mathfrak{n}^{\pi,\theta}$ denote the expected reward of policy $\theta \in \Theta_\mathfrak{n}^\pi$ in the nature MDP $M_\mathfrak{n}^\pi$. Then, by construction, the following holds:

**Lemma 1.** *Given an agent policy $\pi \in \Pi$ and nature policy $\theta \in \Theta_\mathfrak{n}^\pi$. Then, the value of $\theta$ in the nature MDP equals the value of $\theta$ against $\pi$ in the RPOMDP, i.e., $V_\mathfrak{n}^{\pi,\theta} = V^{\pi,\theta}$.*

**Corollary 1.** *There exists a best-response nature policy $\theta_\pi^* \in \mathrm{argmin}_{\theta \in \Theta_\mathfrak{n}^\pi} V_\mathfrak{n}^{\pi,\theta}$.*

Corollary 1 follows from the observation that valid policies in $M_\mathfrak{n}^\pi$ exactly overlap with the set of policies described in Theorem 3. This implies we can find $\theta_\pi^*$ by solving, i.e., finding the optimal policy for, the nature MDP $M_\mathfrak{n}^\pi$. The construction of the nature MDP generalizes the robust Markov chain construction of Galesloot et al. [15] to arbitrary memorization schemes $\tau$ of the agent. We note that $A_\mathfrak{n}$ is finite if $\mathcal{P}$ is a convex polytope[4], and $S_\mathfrak{n}$ is discrete if $X$ is discrete. Both conditions hold for the setting of Galesloot et al. [15], which is why their robust policy evaluation scales to relatively large state spaces. However, the state-space of $M_\mathfrak{n}^\pi$ is continuous for belief-based policies, the action space is continuous in general, and the resulting value function of $M_\mathfrak{n}^\pi$ can be discontinuous and non-convex. Thus, in practice, we may have to resort to an off-the-shelf approximation method for solving $M_\mathfrak{n}^\pi$, incurring an approximation error. We note that the quality of the approximation and the scalability of solving the nature MDP directly connect to advances in solving continuous-state MDPs.

---

[4]This is commonly the case if the uncertainty set is comprised of probability intervals or the $\ell_1$ norm, but generally not for uncertainty sets based on the $\ell_2$ norm or the Kullback-Leibler divergence.

# 5 Efficient Approximations for Robust Agent Policies

In this section, we lift two value approximation methods from POMDPs to their robust counterparts in RPOMDP. These approximations are helpful in multiple ways. Many existing POMDP algorithms, both online [59] and offline [52], use approximations as upper bounds to guide exploration and initialize value estimates. While any choice for $\theta \in \Theta$ is a valid upper bound, using tighter upper bounds often leads to better results more quickly [30]. Furthermore, these approximations are ideal candidates to serve as baselines and provide sanity checks in an evaluation, such as the one we conduct in Section 6, as they provide reasonable bounds on performance that are efficient to compute.

Recall $b \in \Delta(S)$ is an agent belief. Let $\mathfrak{b}_s$ be the *unit belief* with $b(s) = 1$, and $\mathcal{B}_S = \{\mathfrak{b}_s \mid s \in S\} \subset \Delta(S)$ be the set of all such beliefs. Then, we define robust variants of the QMDP-bound [34] and the *fast informed bound* [FIB; 21], namely *robust QMDP* (RQMDP) and *robust FIB* (RFIB), as:

**Definition 4.** $Q_{RMDP}$ and $Q_{RFIB}$ are the fixed point of the operators $H_{RQMDP}$ and $H_{RFIB}$, defined as:

$$H_{RQMDP}Q(b,a) = \sum_{s \in S} b(s) \Big[ R(s,a) + \gamma \inf_{u \in \mathcal{U}} \sum_{s' \in S} \mathcal{T}(u)(s' \mid s, a) \max_{a' \in A} Q(\mathfrak{b}_{s'}, a') \Big], \text{ and} \quad (2)$$

$$H_{RFIB}Q(b,a) = \sum_{s \in S} b(s) \Big[ R(s,a) + \gamma \inf_{u \in \mathcal{U}} \sum_{o \in \Omega} \max_{a' \in A} \sum_{s' \in S} \mathcal{P}(u)(s', o \mid s, a) Q(\mathfrak{b}_{s'}, a') \Big]. \quad (3)$$

We prove both these operators are contraction mappings in Appendix C of the supplemental material, guaranteeing the existence and uniqueness of these fixed points. Intuitively, RFIB corresponds to the worst-case value under the assumption that the agent observes the current and future states from the next timestep onwards with a one-step delay, while RQMDP corresponds to the same assumption with no delay. This matches their POMDP variants. Let $Q_{\text{RPOMDP}}$ be the fixed point of the robust Bellman equation for RPOMDPs [45]. We highlight the following property:

**Theorem 4.** *Regarding tightness, the following inequalities on the fixed points hold:*

$$\forall b \in \Delta(S), \forall a \in A \colon Q_{RMDP}(b,a) \geq Q_{RFIB}(b,a) \geq Q_{RPOMDP}(b,a).$$

We provide a proof in Appendix C of the supplemental material. Intuitively, the theorem follows from the fact that the operators are contraction mappings and unequal. As for their non-robust variants, both $Q_{\text{RMDP}}$ and $Q_{\text{RFIB}}$ depend only on the $Q$-values for state-action pairs. Thus, precomputing the (approximate) fixed point for beliefs $\mathfrak{b}_s \in \mathcal{B}_S$ allows computing the bounds for any belief $b$ efficiently. Whether or not this precomputation is tractable depends on the uncertainty sets. Both are convex optimization problems for convex uncertainty sets. In particular, the inner supremum of Equation (2) is solvable with a linear program, or by using an efficient *bisection method* [43, Section 7.2] if the uncertainty sets are convex polytopes, while for Equation (3) it is solvable via a linear relaxation of a mixed-integer program, which means both require polynomial time.

# 6 Experimental Evaluation

We empirically evaluate our benchmarks and evaluation pipeline via the following questions:

- **(Q1) Evaluation pipeline.** How accurate is our pipeline in evaluating policies?
- **(Q2) Benchmarks.** Can our proposed benchmarks be solved using a naive nature heuristic, or one of our efficient approximations? How does this compare to other benchmarks?
- **(Q3) Approximations.** How do our approximations perform as baselines?

We first provide a brief overview of our experimental setup, and then address **(Q1)-(Q3)**. We include more details in Appendix D, and provide the code to reproduce the experiments on Zenodo [29].

## 6.1 Experimental Setup

**Implementation and algorithms.** We implement our evaluation method in the Julia programming language, using a variant of the `POMDPs.jl` framework [12] for RPOMDPs with interval uncertainty sets. To solve these RPOMDPs, we use three separate algorithms: a variant of RHSVI [45] — with minor alterations, as described in Appendix D.1 — as well as our approximate solvers RQMDP and RFIB from Section 5. With RHSVI, we compute both robust policies on the RPOMDP $\mathcal{M}$, as well as

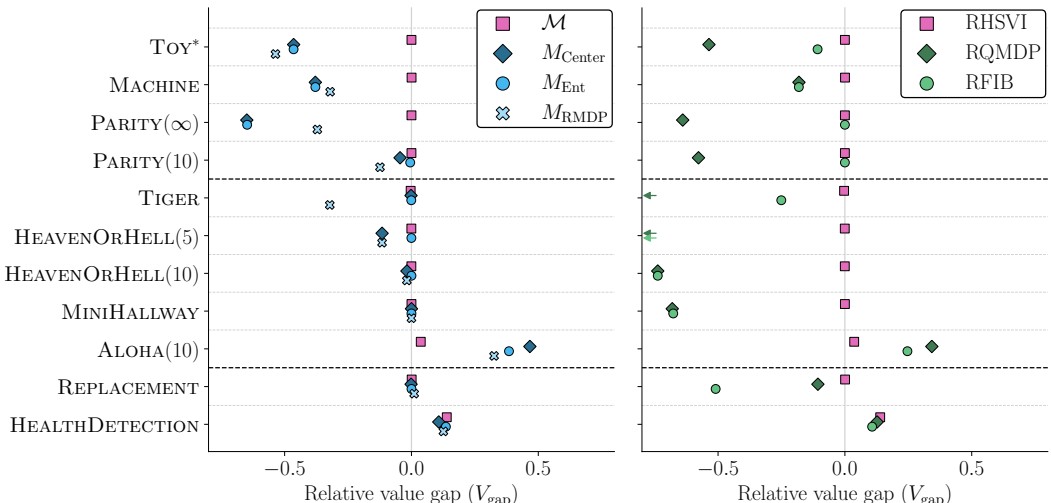

Figure 4: The relative value gap $V_{\text{gap}}(\pi) = V^{\pi,\theta_\pi^*} - \tilde{V}/|\tilde{V}|$, for each policy $\pi$, where $V^{\pi,\theta_\pi^*}$ is obtained by (approximately) solving the nature MDP $M_{\mathfrak{n}}^\pi$. On the left, policies $\pi$ are computed using RHSVI on the RPOMDP model $\mathcal{M}$ or on the naive POMDP models $M_{\text{Center}}$, $M_{\text{Ent}}$, and $M_{\text{RMDP}}$. On the right, policies $\pi$ are computed using different RPOMDP algorithms. Arrows denote outliers.

non-robust POMDP policies for the simplified models $M_{\text{Center}}$, $M_{\text{Ent}}$ and $M_{\text{RMDP}}$, which correspond to the naive nature policies $\theta_{\text{Center}}, \theta_{\text{Ent}}, \theta_{\text{RMDP}}$ in Section 3. Given an agent policy, we construct a nature MDP in the POMDPs.jl framework, and solve it using the native Julia implementation of Monte Carlo tree search (MCTS) for continuous-state MDPs [8]. For evaluation, we run MCTS five times and report the lowest value; unnormalized values and standard deviation are in Appendix D.2.

**Metric.** We test our evaluation pipeline on the abovementioned algorithms to compute the value $V^{\pi,\theta_\pi^*}$ for each policy $\pi$ by solving the nature MDP. From these policy evaluations, we compute a *relative value gap*: $V_{\text{gap}}(\pi) = \frac{V^{\pi,\theta_\pi^*} - \tilde{V}}{|\tilde{V}|}$, where $\tilde{V}$ denotes the (approximate) value of the RPOMDP $\mathcal{M}$ computed using RHSVI. A negative value gap indicates that our pipeline has determined a policy is suboptimal. In contrast, a positive value gap means our pipeline could not compute the worst-case value, i.e., due to the error incurred from the approximation method used to solve the nature MDP.

**Benchmarks.** We test our evaluation pipeline on three sets of benchmarks. Firstly, we use our novel benchmarks as introduced in Section 3: TOY* and ECHO, as well as the finite- and infinite chain environments PARITY(10) and PARITY($\infty$). Secondly, we lift several POMDPs from the literature into RPOMDPs: TIGER [6], MINIHALLWAY [33], and ALOHA [24], as well as an expanded variant of HEAVENORHELL [4] (also used in [45]). We construct these RPOMDPs such that for any $\mathcal{T}(u)$, any transition probability is less than 0.5 times higher or lower than the nominal POMDP, with no alterations to the observation function. Lastly, we add partial observability to two benchmarks from the RMDP literature: HEALTHDETECTION [16] and REPLACEMENT [10]. The former can be interpreted as an RPOMDP with little changes. For the latter, we add partial observability by adding *measuring actions* that reveal the state at the cost of incurring a negative reward [28]. Appendix D.2 provides detailed descriptions of the environments and their dimensions.

## 6.2 Results & Analysis

We aggregate our results in two separate plots (Figure 4), which we analyze below:

**(Q1): Our evaluation pipeline is accurate for smaller environments.** For the smaller environments (MINIHALLWAY and above), our evaluation of RHSVI achieves a value gap close to zero, and the value gaps for all approximation methods are less than or equal to zero. Thus, for these environments, our evaluation pipeline is likely accurate. However, for larger environments (ALOHA and HEALTHDETECTION), some policies achieve positive value gaps. In these cases, MCTS fails to find an accurate solution for the nature MDP, leading to inaccurate policy evaluation.

**(Q2): Our novel benchmarks cannot be solved naively.** For TOY*, ECHO, and PARITY($\infty$), all the policies computed with naive nature heuristics have significant value gaps. Moreover, while RFIB yields an optimal policy for both variants of PARITY, it does not solve TOY* or ECHO, despite their relatively small state spaces. In contrast, for all other benchmarks, at least one policy computed from a naive nature heuristic performs on par with the policy computed with RHSVI on the RPOMDP $\mathcal{M}$.

**(Q3): RFIB is an adequate baseline.** RFIB outperforms all naive nature heuristics for TOY*, ECHO, and both variants of PARITY, on which it also is optimal. This confirms RFIB is useful as a computationally inexpensive baseline for RPOMDPs. However, as we see for the other benchmarks, RFIB is not guaranteed to perform well on all benchmarks. In contrast to RFIB, RQMDP performs worse in general, but is still a valuable baseline for models where RFIB is too expensive to compute.

# 7   Related Work

**RPOMDP evaluation.** No prior work exists that explicitly tackles the evaluation problem for RPOMDPs. However, methods that aim to solve RPOMDPs often make implicit assumptions about evaluation. Most notably, Galesloot et al. [15] proposes a pessimistic iterative planning framework that uses an evaluation method similar to the one used here. However, this work focuses on finding and evaluating finite state controllers for stationary nature policies only, while our evaluation method is more generic. More broadly, RPOMDP solvers exist that compute belief-based policies [45], history-based policies [22, 41] or policies represented as finite state controllers [9, 15]. In all these works, the benchmarks consist of POMDPs with $\epsilon$-uncertainty around the original transition and observation functions, and include no discussion on why these benchmarks are picked. In particular, such variants of both TIGER and HEAVENORHELL have been used as RPOMDP benchmarks [22, 41, 45], while our work shows that both are naively solvable.

**Related settings.** Prior work has evaluated policies by sampling POMDPs from the uncertainty set [5], which, in contrast to our evaluation method, does not guarantee finding the worst-case value. Furthermore, related settings include policy optimization and evaluation for finite sets of POMDPs, where model uncertainty is static and not rectangular [14], value iteration with side information under distributional robustness [40], and value iteration under varying pessimism levels [49]. Alternative settings include robustifying POMDP policies against observation perturbations [7] and robust active measuring [28]. Lastly, we note that solver evaluation has been studied in different fields, including for MDPs [20] and reinforcement learning, both in general [44, 18] and for RMDPs in particular [60].

# 8   Conclusion & Discussion

In this paper, we consider three understudied components of the RPOMDP evaluation pipeline: (1) identifying suitable benchmarks, (2) robust policy evaluation, and (3) efficient baseline algorithms. We introduce novel methods to tackle all three problems, and empirically confirm that the resulting pipeline is sound. Future work could use our approximations to guide RPOMDP solvers, or introduce more specific approximations of the nature MDP to increase scalability.

**Limitations.** As shown in section 6, approximation errors in solving the nature MDP may result in incorrect values. Thus, care should be taken in deciding which approximation method to use and assessing the resulting values. Furthermore, our methods and analysis in this paper are restricted to $\langle s, a \rangle$-rectangular and convex uncertainty sets. Yet, these are common assumptions for RPOMDPs [45, 15], and any non-rectangular problem can be overapproximated (conservatively) by assuming rectangularity. We note that identifying less conservative cases of rectangularity while maintaining tractability is still an open problem in RMDPs [58, 17].

**Practical recommendations.** For future research on RPOMDPs, we have two practical recommendations. Firstly, RPOMDP solvers should be tested on benchmarks that are not naively solvable, which can be tested theoretically (as done in Section 3) or empirically (as done in Section 6). Secondly, RPOMDP solvers should be evaluated using a robust policy evaluation method that finds the worst-case value from a full range of possible nature policies, such as the one proposed in Section 4.

## Acknowledgments and Disclosure of Funding

We would like to thank the anonymous reviewers for their useful comments. This work has been partially funded by the ERC Starting Grant DEUCE (101077178).

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

# A Solvability Classes and Benchmark Models

This section contains all proofs related to Section 3 of the main paper.

## A.1 Solvability proofs

First, we show the correctness of the policies used in section Section 3 to explain the solvability classes. We restate the RPOMDP used in Section 3.1.

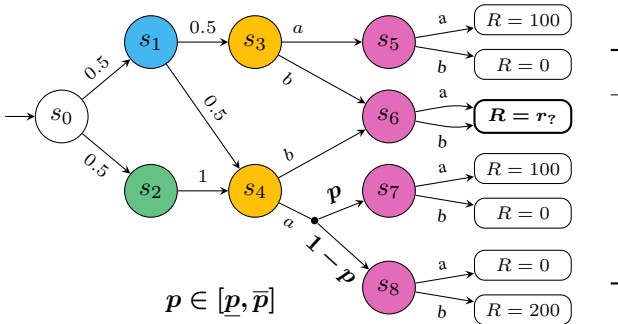

| Solvability | $\underline{p}$ | $\overline{p}$ | $r_?$ |
|---|---|---|---|
| Trivially | 0.2 | 0.6 | 80 |
| Center | 0.1 | 0.6 | 80 |
| Entropy | 0.1 | 0.6 | 80 |
| RMDP | 0.1 | 0.6 | 80 |
| Stationary | 0.1 | 0.9 | 80 |
| None of the above | 0.1 | 0.9 | 70 |

Figure 5: Example RPOMDP with parameter values for solvability classes. We use TOY$^*$ to denote the model with the parameter values from the bottom line.

Let $h^{\bullet\bullet} = \circ\bullet\bullet\bullet$, $h^{\bullet\bullet} = \circ\bullet\bullet\bullet$, $h^{\bullet\bullet\bullet} = \bullet\bullet\bullet a\bullet$, and $h^{\bullet\bullet\bullet} = \circ\bullet\bullet a\bullet$ denote the histories for which the agent or nature need to make non-singleton choices. Given agent and nature policies $\pi$ and $\theta$, the value function of the RPOMDP in fig. 5 can be expressed by the following function:

$$
\begin{aligned}
V^{\pi,\theta} = {} & 0.5 \cdot 0.5 \cdot \pi(h^{\bullet\bullet})(a) \cdot \pi(h^{\bullet\bullet\bullet})(a) \cdot 100 \\
& + 0.5 \cdot 0.5 \cdot \pi(h^{\bullet\bullet})(b) \cdot r_? \\
& + 0.5 \cdot 0.5 \cdot \pi(h^{\bullet\bullet})(b) \cdot r_? \\
& + 0.5 \cdot 0.5 \cdot \pi(h^{\bullet\bullet})(a) \cdot \theta(h^{\bullet\bullet}, a)(p) \cdot \pi(h^{\bullet\bullet\bullet})(a) \cdot 100 \\
& + 0.5 \cdot 0.5 \cdot \pi(h^{\bullet\bullet})(a) \cdot (1 - \theta(h^{\bullet\bullet}, a)(p)) \cdot \pi(h^{\bullet\bullet\bullet})(b) \cdot 200 \\
& + 0.5 \cdot \pi(h^{\bullet\bullet})(b) \cdot r_? \\
& + 0.5 \cdot \pi(h^{\bullet\bullet})(a) \cdot \theta(h^{\bullet\bullet}, a)(p) \cdot \pi(h^{\bullet\bullet\bullet})(a) \cdot 100 \\
& + 0.5 \cdot \pi(h^{\bullet\bullet})(a) \cdot (1 - \theta(h^{\bullet\bullet}, a)(p)) \cdot \pi(h^{\bullet\bullet\bullet})(b) \cdot 200
\end{aligned}
$$

We simplify and rewrite this function to remove the $b$ terms:

$$
\begin{aligned}
= {} & 25 \cdot \pi(h^{\bullet\bullet})(a) \cdot \pi(h^{\bullet\bullet\bullet})(a) \\
& + 0.5 \cdot r_? \cdot (1 - \pi(h^{\bullet\bullet})(a)) \\
& + 25 \cdot \pi(h^{\bullet\bullet})(a) \cdot \theta(h^{\bullet\bullet}, a)(p) \cdot \pi(h^{\bullet\bullet\bullet})(a) \\
& + 50 \cdot \pi(h^{\bullet\bullet})(a) \cdot (1 - \theta(h^{\bullet\bullet}, a)(p)) \cdot (1 - \pi(h^{\bullet\bullet\bullet})(a)) \\
& + 0.5 \cdot r_? \cdot (1 - \pi(h^{\bullet\bullet})(a)) \\
& + 50 \cdot \pi(h^{\bullet\bullet})(a) \cdot \theta(h^{\bullet\bullet}, a)(p) \cdot \pi(h^{\bullet\bullet\bullet})(a) \\
& + 100 \cdot \pi(h^{\bullet\bullet})(a) \cdot (1 - \theta(h^{\bullet\bullet}, a)(p)) \cdot (1 - \pi(h^{\bullet\bullet\bullet})(a))
\end{aligned}
$$

We can now further simplify the notation with $a^{🔵} = \pi(h^{🟡🟢})(a), a^{🟢} = \pi(h^{🟡🟠})(a), a^{🔵🟢} = \pi(h^{🟡🟢🟢})(a), a^{🟢🟢} = \pi(h^{🟡🟠🟢})(a)$ and $p^{🔵} = \theta(h^{🟡🟢}, a)(p), p^{🟢} = \theta(h^{🟡🟠}, a)(p)$.

$$
\begin{aligned}
&= 25a^{🔵}a^{🔵🟢} \\
&\quad + 0.5r_?(1 - a^{🔵}) \\
&\quad + 25a^{🔵}p^{🔵}a^{🔵🟢} \\
&\quad + 50a^{🔵}(1 - p^{🔵})(1 - a^{🔵🟢}) \\
&\quad + 0.5r_?(1 - a^{🟢}) \\
&\quad + 50a^{🟢}p^{🟢}a^{🟢🟢} \\
&\quad + 100a^{🟢}(1 - p^{🟢})(1 - a^{🟢🟢}) \\
&= 25a^{🔵}a^{🔵🟢} \\
&\quad + 0.5r_? - 0.5r_?a^{🔵} \\
&\quad + 25a^{🔵}p^{🔵}a^{🔵🟢} \\
&\quad + 50a^{🔵} - 50a^{🔵}p^{🔵} - 50a^{🔵}a^{🔵🟢} + 50a^{🔵}p^{🔵}a^{🔵🟢} \\
&\quad + 0.5r_? - 0.5r_?a^{🟢} \\
&\quad + 50a^{🟢}p^{🟢}a^{🟢🟢} \\
&\quad + 100a^{🟢} - 100a^{🟢}p^{🟢} - 100a^{🟢}a^{🟢🟢} + 100a^{🟢}p^{🟢}a^{🟢🟢} \\
&= r_? + (50 - 0.5r_?)a^{🔵} - 50a^{🔵}p^{🔵} - 25a^{🔵}a^{🔵🟢} + 75a^{🔵}p^{🔵}a^{🔵🟢} \\
&\quad + (100 - 0.5r_?)a^{🟢} - 100a^{🟢}p^{🟢} - 100a^{🟢}a^{🟢🟢} + 150a^{🟢}p^{🟢}a^{🟢🟢}
\end{aligned}
$$

Recall the optimal value for the agent $V^{🅰} = \max_{\pi \in \Pi} \min_{\theta \in \Theta} V^{\pi,\theta}$ and nature $V^{🅽} = \min_{\theta \in \Theta} \max_{\pi \in \Pi} V^{\pi,\theta}$ with corresponding optimal agent and nature policies $\pi^*$ and $\theta^*$. If $V^{🅰} = V^{🅽}$, then the underlying game has a *Nash equilibrium* [46] in which case $\pi^*$ and $\theta^*$ are called *Nash policies*. We can therefore show that an agent policy is optimal if it can guarantee the same value as a nature policy can guarantee, and vice versa. Using this logic, we consider the following two agent policies.

First, the *safe* policy $\pi_s^*$ always picks the safe action $b$ in the 🟡 states, so regardless of the previous observation, which yields a guaranteed value of $r_?$. The action in the 🟢 states does not matter after the safe action $b$.

$$
\begin{aligned}
\forall \theta \in \Theta : V^{\pi_s^*,\theta} &= r_? + (50 - 0.5r_?) \cdot 0 - 50 \cdot 0 \cdot p^{🔵} - 25 \cdot 0 \cdot a^{🔵🟢} + 75 \cdot 0 \cdot p^{🔵}a^{🔵🟢} \\
&\quad + (100 - 0.5r_?) \cdot 0 - 100 \cdot 0 \cdot p^{🟢} - 100 \cdot 0 \cdot a^{🟢🟢} + 150 \cdot 0 \cdot p^{🟢}a^{🟢🟢} \\
&= r_? + 0 - 0 - 0 + 0 + 0 - 0 - 0 + 0 \\
&= r_?
\end{aligned}
$$

Next, the *risky* policy $\pi_r^*$ picks the riskier action $a$ in these 🟡 states if it's previous observation was 🟢, and the safe action $b$ otherwise. Furthermore, $\pi_r^*$ picks action $b$ in the 🟡 states. The value that $\pi_r^*$ can guarantee depends on the bounds on $p$.

$$
\begin{aligned}
\forall \theta \in \Theta : V^{\pi_r^*,\theta} &= r_? + (50 - 0.5r_?) \cdot 0 - 50 \cdot 0 \cdot p^{🔵} - 25 \cdot 0 \cdot a^{🔵🟢} + 75 \cdot 0 \cdot p^{🔵}a^{🔵🟢} \\
&\quad + (100 - 0.5r_?) \cdot 1 - 100 \cdot 1 \cdot p^{🟢} - 100 \cdot 1 \cdot 0 + 150 \cdot 1 \cdot p^{🟢} \cdot 0 \\
&= r_? + 0 - 0 - 0 + 0 + 100 - 0.5r_? - 100p^{🟢} - 0 + 0 \\
&= 0.5r_? + 100 - 100p^{🟢}
\end{aligned}
$$

Similarly, we can look at the value certain nature policies can guarantee.

We find that nature can ensure a value of at most $r_?$ when choosing $p^{🔵}$ and $p^{🟢}$ within certain intervals.

$$
\begin{aligned}
\forall \pi \in \Pi : V^{\pi,\theta} &= r_? + (50 - 0.5r_?)a^{🔵} - 50a^{🔵}p^{🔵} - 25a^{🔵}a^{🔵🟢} + 75a^{🔵}p^{🔵}a^{🔵🟢} \\
&\quad + (100 - 0.5r_?)a^{🟢} - 100a^{🟢}p^{🟢} - 100a^{🟢}a^{🟢🟢} + 150a^{🟢}p^{🟢}a^{🟢🟢}
\end{aligned}
$$

$a^{🟢}, a^{🟢🟢}$, and $p^{🟢}$ are independent of $a^{🔵}, a^{🔵🟢}$, and $p^{🔵}$, so $\forall \pi \in \Pi : V^{\pi,\theta} \leq r_?$ iff:

$$
\forall \pi \in \Pi : (50 - 0.5r_?)a^{🔵} - 50a^{🔵}p^{🔵} - 25a^{🔵}a^{🔵🟢} + 75a^{🔵}p^{🔵}a^{🔵🟢} \leq 0
$$

and
$$\forall \pi \in \Pi : (100 - 0.5r_?)a^{\color{green}\bullet} - 100a^{\color{green}\bullet}p^{\color{green}\bullet} - 100a^{\color{green}\bullet}a^{\color{green}\bullet\bullet} + 150a^{\color{green}\bullet}p^{\color{green}\bullet}a^{\color{green}\bullet\bullet} \leq 0$$

Given arbitrary $a^{\color{green}\bullet}, a^{\color{green}\bullet\bullet} \in [0,1]$, we first compute the interval for $p^{\color{magenta}\bullet}$:
$$(50 - 0.5r_?)a^{\color{green}\bullet} - 50a^{\color{green}\bullet}p^{\color{magenta}\bullet} - 25a^{\color{green}\bullet}a^{\color{green}\bullet\bullet} + 75a^{\color{green}\bullet}p^{\color{magenta}\bullet}a^{\color{green}\bullet\bullet} \leq 0$$
$$a^{\color{green}\bullet}(50 - 0.5r_? - 50p^{\color{magenta}\bullet} - 25a^{\color{green}\bullet\bullet} + 75p^{\color{magenta}\bullet}a^{\color{green}\bullet\bullet}) \leq 0$$

$a^{\color{green}\bullet} \in [0,1]$, so the inequality above holds for all $\pi \in \Pi$ iff:
$$50 - 0.5r_? - 50p^{\color{magenta}\bullet} - 25a^{\color{green}\bullet\bullet} + 75p^{\color{magenta}\bullet}a^{\color{green}\bullet\bullet} \leq 0$$
$$50 - 0.5r_? - 50p^{\color{magenta}\bullet} \leq 25a^{\color{green}\bullet\bullet} - 75p^{\color{magenta}\bullet}a^{\color{green}\bullet\bullet}$$
$$50 - 0.5r_? - 50p^{\color{magenta}\bullet} \leq a^{\color{green}\bullet\bullet}(25 - 75p^{\color{magenta}\bullet})$$

$a^{\color{green}\bullet\bullet} \in [0,1]$, so the inequality above holds for all $\pi \in \Pi$ iff:
$$50 - 0.5r_? - 50p^{\color{magenta}\bullet} \leq 0 \quad \wedge \quad 50 - 0.5r_? - 50p^{\color{magenta}\bullet} \leq 25 - 75p^{\color{magenta}\bullet}$$
$$50 - 0.5r_? \leq 50p^{\color{magenta}\bullet} \quad \wedge \quad 25p^{\color{magenta}\bullet} \leq 0.5r_? - 25$$
$$1 - 0.01r_? \leq p^{\color{magenta}\bullet} \quad \wedge \quad p^{\color{magenta}\bullet} \leq 0.02r_? - 1$$

So we get our first interval $p^{\color{magenta}\bullet} \in [1 - 0.01r_?, 0.02r_? - 1]$. Given arbitrary $a^{\color{green}\bullet}, a^{\color{green}\bullet\bullet} \in [0,1]$, we continue with the interval for $p^{\color{green}\bullet}$:
$$(100 - 0.5r_?)a^{\color{green}\bullet} - 100a^{\color{green}\bullet}p^{\color{green}\bullet} - 100a^{\color{green}\bullet}a^{\color{green}\bullet\bullet} + 150a^{\color{green}\bullet}p^{\color{green}\bullet}a^{\color{green}\bullet\bullet} \leq 0$$
$$a^{\color{green}\bullet}(100 - 0.5r_? - 100p^{\color{green}\bullet} - 100a^{\color{green}\bullet\bullet} + 150p^{\color{green}\bullet}a^{\color{green}\bullet\bullet}) \leq 0$$

$a^{\color{green}\bullet} \in [0,1]$, so the inequality above holds for all $\pi \in \Pi$ iff:
$$100 - 0.5r_? - 100p^{\color{green}\bullet} - 100a^{\color{green}\bullet\bullet} + 150p^{\color{green}\bullet}a^{\color{green}\bullet\bullet} \leq 0$$
$$100 - 0.5r_? - 100p^{\color{green}\bullet} \leq 100a^{\color{green}\bullet\bullet} - 150p^{\color{green}\bullet}a^{\color{green}\bullet\bullet}$$
$$100 - 0.5r_? - 100p^{\color{green}\bullet} \leq a^{\color{green}\bullet\bullet}(100 - 150p^{\color{green}\bullet})$$

$a^{\color{green}\bullet\bullet} \in [0,1]$, so the inequality above holds for all $\pi \in \Pi$ iff:
$$100 - 0.5r_? - 100p^{\color{green}\bullet} \leq 0 \quad \wedge \quad 100 - 0.5r_? - 100p^{\color{green}\bullet} \leq 100 - 150p^{\color{green}\bullet}$$
$$100 - 0.5r_? \leq 100p^{\color{green}\bullet} \quad \wedge \quad 50p^{\color{green}\bullet} \leq 0.5r_?$$
$$1 - 0.005r_? \leq p^{\color{green}\bullet} \quad \wedge \quad p^{\color{green}\bullet} \leq 0.01r_?$$

So we get our second interval $p^{\color{green}\bullet} \in [1 - 0.005r_?, 0.01r_?]$. We can conclude that $\forall \pi \in \Pi : V^{\pi,\theta} \leq r_? \iff p^{\color{magenta}\bullet} \in [1 - 0.01r_?, 0.02r_? - 1] \wedge p^{\color{green}\bullet} \in [1 - 0.005r_?, 0.01r_?]$. We consider two values for $r_?$ in our example, i. e., $70$ and $80$, so we get the following intervals for $p^{\color{magenta}\bullet}$ and $p^{\color{green}\bullet}$:
$$r_? = 70 : p^{\color{magenta}\bullet} \in [0.3, 0.4], p^{\color{green}\bullet} \in [0.65, 0.7]$$
$$r_? = 80 : p^{\color{magenta}\bullet} \in [0.2, 0.6], p^{\color{green}\bullet} \in [0.6, 0.8]$$

### A.1.1 Trivially solvable

Since the trivially solvable model in fig. 5 has $r_? = 80$ and bounds $p \in [0.2, 0.6]$, we know that nature can guarantee a value of $80$ by playing $p^{\color{magenta}\bullet} = p^{\color{green}\bullet} = 0.6$. We therefore know that any agent policy that can guarentee a value of $80$ is optimal as well. In particular, we know that $\pi_r^*$ is optimal, since:

$$\forall \theta \in \Theta : V^{\pi_r^*, \theta} = 0.5r_? + 100 - 100p^{\color{green}\bullet}$$
$$= 0.5 \cdot 80 + 100 - 100p^{\color{green}\bullet}$$
$$= 140 - 100p^{\color{green}\bullet}$$
$$\geq 140 - 100 \cdot 0.6$$
$$= 80$$

To show that this model is trivially solvable, we show that $\pi_r^*$ is optimal for any $\theta \in \Theta$. In other words, for all nature policies in the set of nature policies, there is no agent policy $\pi'$ that achieves a higher reward than $\pi_r^*$ against that particular nature policies.

Given an arbitrary $\theta \in \Theta$, we construct the optimal agent policy $\pi'$ by maximizing over the value function:

$$
\begin{aligned}
\pi' &= \operatorname*{argmax}_{\pi \in \Pi} V^{\pi,\theta} \\
&= \operatorname*{argmax}_{\pi \in \Pi} \Big( r_? + (50 - 0.5r_?)a^{\circ} - 50a^{\circ}p^{\circ} - 25a^{\circ}a^{\circ\circ} + 75a^{\circ}p^{\circ}a^{\circ\circ} \\
&\qquad\qquad + (100 - 0.5r_?)a^{\circ} - 100a^{\circ}p^{\circ} - 100a^{\circ}a^{\circ\circ} + 150a^{\circ}p^{\circ}a^{\circ\circ} \Big) \\
&= \operatorname*{argmax}_{\pi \in \Pi} \Big( 80 + (50 - 0.5 \cdot 80)a^{\circ} - 50a^{\circ}p^{\circ} - 25a^{\circ}a^{\circ\circ} + 75a^{\circ}p^{\circ}a^{\circ\circ} \\
&\qquad\qquad + (100 - 0.5 \cdot 80)a^{\circ} - 100a^{\circ}p^{\circ} - 100a^{\circ}a^{\circ\circ} + 150a^{\circ}p^{\circ}a^{\circ\circ} \Big) \\
&= \operatorname*{argmax}_{\pi \in \Pi} \Big( 80 + 10a^{\circ} - 50a^{\circ}p^{\circ} - 25a^{\circ}a^{\circ\circ} + 75a^{\circ}p^{\circ}a^{\circ\circ} \\
&\qquad\qquad + 60a^{\circ} - 100a^{\circ}p^{\circ} - 100a^{\circ}a^{\circ\circ} + 150a^{\circ}p^{\circ}a^{\circ\circ} \Big) \\
&= \operatorname*{argmax}_{\pi \in \Pi} \Big( 10a^{\circ} - 50a^{\circ}p^{\circ} - 25a^{\circ}a^{\circ\circ} + 75a^{\circ}p^{\circ}a^{\circ\circ} \\
&\qquad\qquad + 60a^{\circ} - 100a^{\circ}p^{\circ} - 100a^{\circ}a^{\circ\circ} + 150a^{\circ}p^{\circ}a^{\circ\circ} \Big)
\end{aligned}
$$

$a^{\circ}, a^{\circ\circ}$, and $p^{\circ}$ are independent of $a^{\circ}, a^{\circ\circ}$, and $p^{\circ}$, so we can compute two parts of the agent policy seprately:

$$
\begin{aligned}
&= \operatorname*{argmax}_{a^{\circ},a^{\circ\circ}} \Big( 10a^{\circ} - 50a^{\circ}p^{\circ} - 25a^{\circ}a^{\circ\circ} + 75a^{\circ}p^{\circ}a^{\circ\circ} \Big) \\
&\quad \times \operatorname*{argmax}_{a^{\circ},a^{\circ\circ}} \Big( 60a^{\circ} - 100a^{\circ}p^{\circ} - 100a^{\circ}a^{\circ\circ} + 150a^{\circ}p^{\circ}a^{\circ\circ} \Big)
\end{aligned}
$$

We begin with $a^{\circ}$ and $a^{\circ\circ}$:

$$
\operatorname*{argmax}_{a^{\circ},a^{\circ\circ}} \Big( 10a^{\circ} - 50a^{\circ}p^{\circ} - 25a^{\circ}a^{\circ\circ} + 75a^{\circ}p^{\circ}a^{\circ\circ} \Big) = \operatorname*{argmax}_{a^{\circ},a^{\circ\circ}} \Big( a^{\circ}(10 - 50p^{\circ} - a^{\circ\circ}(25 - 75p^{\circ})) \Big)
$$

As we are maximizing, we know that $a^{\circ}$ should be 1 if $10 - 50p^{\circ} - a^{\circ\circ}(25 - 75p^{\circ}) > 0$ and we can set it to 0 otherwise. We show $10 - 50p^{\circ} - a^{\circ\circ}(25 - 75p^{\circ}) \leq 0$ regardless of $a^{\circ\circ}$:

$$
\begin{aligned}
10 - 50p^{\circ} - a^{\circ\circ}(25 - 75p^{\circ}) &\leq 0 \\
10 - 50p^{\circ} &\leq a^{\circ\circ}(25 - 75p^{\circ})
\end{aligned}
$$

$a^{\circ\circ} \in [0, 1]$, so the inequality above holds for all $a^{\circ\circ}$ iff:

$$
\begin{aligned}
10 - 50p^{\circ} \leq 0 &\quad \wedge \quad 10 - 50p^{\circ} \leq 25 - 75p^{\circ} \\
10 \leq 50p^{\circ} &\quad \wedge \quad 25p^{\circ} \leq 15 \\
0.2 \leq p^{\circ} &\quad \wedge \quad p^{\circ} \leq 0.6
\end{aligned}
$$

Since $\forall \theta \in \Theta : p^{\circ} \in [0.2, 0.6]$, we have that choosing $a^{\circ} = 0$ is optimal.

We continue with $a^{\circ}$ and $a^{\circ\circ}$:

$$
\operatorname*{argmax}_{a^{\circ},a^{\circ\circ}} \Big( 60a^{\circ} - 100a^{\circ}p^{\circ} - 100a^{\circ}a^{\circ\circ} + 150a^{\circ}p^{\circ}a^{\circ\circ} \Big) = \operatorname*{argmax}_{a^{\circ},a^{\circ\circ}} \Big( a^{\circ}(60 - 100p^{\circ} - a^{\circ\circ}(100 - 150p^{\circ})) \Big)
$$

As we are maximizing, we know that $a^{\circ}$ should be 1 if $60 - 100p^{\circ} - a^{\circ\circ}(100 - 150p^{\circ}) \geq 0$ and we can set it to 0 otherwise. We know $60 - 100p^{\circ} \geq 0$ regardless of $p^{\circ}$:

$$
\begin{aligned}
60 - 100p^{\circ} &\geq 60 - 100 \cdot 0.6 \\
&= 0
\end{aligned}
$$

Since we can set $a^{\circ\circ}$ to 0, we already know that $60 - 100p^{\circ} - a^{\circ\circ}(100 - 150p^{\circ}) \geq 0$, so we set $a^{\circ}$ to 1. Finally, we determine the optimal $a^{\circ\circ}$.

$$
\begin{aligned}
a^{\circ\circ}(100 - 150p^{\circ}) &\geq a^{\circ\circ}(100 - 150 \cdot 0.6) \\
&= a^{\circ\circ} \cdot 0 \\
&= 0
\end{aligned}
$$

Since we subtract with $a^{\circ\circ}(100 - 150p^{\circ}) \geq 0$, it is optimal to set $a^{\circ\circ}$ to 0.

The agent policy that we constructed $\pi'$ with $a^{\circ} = 0, a^{\circ} = 1$, and $a^{\circ\circ} = 0$ is optimal against all $\theta \in \Theta$, and this policy is exactly $\pi_r^*$. Since the agent policy we compute against any nature policy $\theta \in \Theta$ is optimal against the entire set of nature polcies $\Theta$, we conclude that our trivially solvable model in fig. 5 is indeed trivially solvable.

### A.1.2 Naively solvable

Like in the trivially solvable model in fig. 5, nature can guarantee a value of 80 in the center, entropy, and RMDP solvable models in fig. 5 by playing $p^{\circ} = p^{\circ} = 0.6$. We therefore also know $\pi_r^*$ is again optimal, since it can guarantee a value of 80.

The naive nature policies in the center, entropy, and RMDP solvable models are:

- $\theta_{\text{Center}}$ assigns $p^{\circ} = p^{\circ} = 0.35$, as Centroid $(p) = 0.35$.
- $\theta_{\text{Ent}}$ assigns $p^{\circ} = p^{\circ} = 0.5$, as this creates the maximum entropy between states $s_7$ and $s_8$, namely $\{s_7 \mapsto 0.5, s_8 \mapsto 0.5\}$.
- $\theta_{\text{RMDP}}$ assigns $p^{\circ} = p^{\circ} = 0.6$, as in the fully observable model, it is always optimal for nature to minimize the chance of reaching state $s_8$, and therefore to maximize $p$.

All three of these naive nature policies are contained in the set of policies of the trivially solvable model, for which we have shown that $\pi_r^*$ is optimal.

Since the agent policies we compute against $\theta_{\text{Center}}, \theta_{\text{Ent}}$, and $\theta_{\text{RMDP}}$ are optimal against the entire set of nature polcies $\Theta$, we conclude that our center, entropy and RMDP solvable models in fig. 5 are indeed center, entropy, and RMDP solvable.

We also note that the center, entropy, and RMDP solvable models in fig. 5 are not trivially solvable. For example, $\pi_r^*$ is not optimal against the nature policy $\theta'$ with $p^{\circ} = p^{\circ} = 0.1$. We again construct the optimal agent policy $\pi'$ by maximizing over the value function:

$$
\begin{aligned}
\pi' &= \underset{\pi \in \Pi}{\operatorname{argmax}} V^{\pi, \theta'} \\
&= \underset{\pi \in \Pi}{\operatorname{argmax}}\Big(80 + 10a^{\circ} - 50a^{\circ} \cdot 0.1 - 25a^{\circ}a^{\circ\circ} + 75a^{\circ} \cdot 0.1 \cdot a^{\circ\circ} \\
&\qquad + 60a^{\circ} - 100a^{\circ} \cdot 0.1 - 100a^{\circ}a^{\circ\circ} + 150a^{\circ} \cdot 0.1 \cdot a^{\circ\circ}\Big) \\
&= \underset{\pi \in \Pi}{\operatorname{argmax}}\Big(80 + 10a^{\circ} - 5a^{\circ} - 25a^{\circ}a^{\circ\circ} + 7.5a^{\circ}a^{\circ\circ} \\
&\qquad + 60a^{\circ} - 10a^{\circ} - 100a^{\circ}a^{\circ\circ} + 15a^{\circ}a^{\circ\circ}\Big) \\
&= \underset{\pi \in \Pi}{\operatorname{argmax}}\Big(80 + 5a^{\circ} - 17.5a^{\circ}a^{\circ\circ} + 50a^{\circ} - 85a^{\circ}a^{\circ\circ}\Big)
\end{aligned}
$$

Since we are maximizing, it is optimal to set $a^{\circ}$ and $a^{\circ}$ to 1 and $a^{\circ\circ}$ and $a^{\circ\circ}$ to 0. This $\pi'$ results in value $V^{\pi', \theta'} = 80 + 5 + 50 = 135$, whereas $\pi_r^*$ results in value $V^{\pi_r^*, \theta'} = 80 + 50 = 130$.

Since the optimal agent policy $\pi'$ for nature policy $\theta' \in \Theta$ is not optimal against the entire set of nature policies $\Theta$, the center, entropy, and RMDP solvable models in fig. 5 are not trivially solvable.

### A.1.3 Stationary solvable

Since the stationary solvable model in fig. 5 has $r_? = 80$ and bounds $p \in [0.1, 0.9]$, we know that nature can guarantee a value of 80 by playing any policy with $p^{\circ} \in [0.2, 0.6]$ and $p^{\circ} \in [0.6, 0.8]$. We therefore know that any agent policy that can guarentee a value of 80 is optimal as well. In particular, we know that $\pi_s^*$ is optimal, since this policy always results in a value of $r_? = 80$.

To show that this model is stationary solvable, we show that $\pi_s^*$ is also optimal against the set of stationary nature policies $\Theta^{\text{Sta}}$. Since $\pi_s^*$ guarantees a value of $80$, this agent policy is optimal if there is stationary nature policy that can also guarentee a value of $80$. We know that the stationary nature policy $p^{\circ} = p^{\circ} = 0.6$ guarantees a value of $80$, so we can conclude that the stationary solvable model in fig. 5 is indeed stationary solvable.

We also note that the stationary solvable model in fig. 5 is not trivially or naively solvable, as the optimal agent policies against the nature policies $\theta_{\text{Center}}, \theta_{\text{Ent}}$, and $\theta_{\text{RMDP}}$ cannot guarantee a value of $80$ against the entire set of nature policies.

The naive policies in the stationary solvable models are:

- $\theta_{\text{Center}}$ assigns $p^{\circ} = p^{\circ} = 0.5$, as Centroid $(p) = 0.5$.
- $\theta_{\text{Ent}}$ assigns $p^{\circ} = p^{\circ} = 0.5$, as this creates the maximum entropy between states $s_7$ and $s_8$, namely $\{s_7 \mapsto 0.5, s_8 \mapsto 0.5\}$.
- $\theta_{\text{RMDP}}$ assigns $p^{\circ} = p^{\circ} = 0.9$, as in the fully observable model, it is always optimal for nature to minimize the chance of reaching state $s_8$, and therefore to maximize $p$.

As shown in appendix A.1.2, we know that $\pi_r^*$ is optimal against $\theta_{\text{Center}}$ and $\theta_{\text{Ent}}$. However, $\pi_r^*$ is not optimal against the entire set of nature policies, as nature can achieve a value $< 80$ when playing a nature policy $\theta'$ with $p^{\circ} > 0.6$:

$$
\begin{aligned}
V^{\pi_r^*,\theta'} &= 0.5 \cdot r_? + 100 - 100 p^{\circ} \\
&= 0.5 \cdot 80 + 100 - 100 p^{\circ} \\
&= 140 - 100 p^{\circ} \\
&< 140 - 100 \cdot 0.6 \\
&= 140 - 60 \\
&= 80
\end{aligned}
$$

Next, we show the stationary solvable model is not RMDP solvable by constructing the optimal agent policy $\pi'$ against $\theta_{\text{RMDP}}$ and showing that this agent policy cannot guarantee a value of $80$ against the entire set of nature policies $\Theta$.

$$
\begin{aligned}
\pi' &= \underset{\pi \in \Pi}{\arg\max}\, V^{\pi,\theta_{\text{RMDP}}} \\
&= \underset{\pi \in \Pi}{\arg\max}\Big(80 + 10 a^{\circ} - 50 a^{\circ} \cdot 0.9 - 25 a^{\circ} a^{\circ\circ} + 75 a^{\circ} \cdot 0.9 \cdot a^{\circ\circ} \\
&\qquad\qquad + 60 a^{\circ} - 100 a^{\circ} \cdot 0.9 - 100 a^{\circ} a^{\circ\circ} + 150 a^{\circ} \cdot 0.9 \cdot a^{\circ\circ}\Big) \\
&= \underset{\pi \in \Pi}{\arg\max}\Big(80 + 10 a^{\circ} - 45 a^{\circ} - 25 a^{\circ} a^{\circ\circ} + 67.5 a^{\circ} a^{\circ\circ} \\
&\qquad\qquad + 60 a^{\circ} - 90 a^{\circ} - 100 a^{\circ} a^{\circ\circ} + 135 a^{\circ} a^{\circ\circ}\Big) \\
&= \underset{\pi \in \Pi}{\arg\max}\Big(80 - 35 a^{\circ} + 42.5 a^{\circ} a^{\circ\circ} + -30 a^{\circ} + 35 a^{\circ} a^{\circ\circ}\Big)
\end{aligned}
$$

Since we are maximizing, it is optimal to set $a^{\circ}, a^{\circ}, a^{\circ\circ}$, and $a^{\circ\circ}$ all to $1$. This $\pi'$ results in value $V^{\pi',\theta_{\text{RMDP}}} = 80 - 35 + 42.5 - 30 + 35 = 92.5$, whereas $\pi_s^*$ results in value $V^{\pi_s^*,\theta_{\text{RMDP}}} = 80$. However, $\pi'$ is not optimal against the entire set of nature policies $\Theta$, as nature can achieve a value of $< 80$

when playing a nature policy $\theta'$ with $p^{\circ} + 2p^{\circ} < 1.8$:

$$
\begin{aligned}
V^{\pi',\theta'} &= 80 + 10a^{\circ} - 50a^{\circ}p^{\circ} - 25a^{\circ}a^{\circ\circ} + 75a^{\circ}p^{\circ}a^{\circ\circ} \\
&\quad + 60a^{\circ} - 100a^{\circ}p^{\circ} - 100a^{\circ}a^{\circ\circ} + 150a^{\circ}p^{\circ}a^{\circ\circ} \\
&= 80 + 10 \cdot 1 - 50 \cdot 1 \cdot p^{\circ} - 25 \cdot 1 \cdot 1 + 75 \cdot 1 \cdot p^{\circ} \cdot 1 \\
&\quad + 60 \cdot 1 - 100 \cdot 1 \cdot p^{\circ} - 100 \cdot 1 \cdot 1 + 150 \cdot 1 \cdot p^{\circ} \cdot 1 \\
&= 80 + 10 - 50p^{\circ} - 25 + 75p^{\circ} + 60 - 100p^{\circ} - 100 + 150p^{\circ} \\
&= 35 + 25p^{\circ} + 50p^{\circ} \\
&= 35 + 25(p^{\circ} + 2p^{\circ}) \\
&< 35 + 25 \cdot 1.8 \\
&= 35 + 45 \\
&= 80
\end{aligned}
$$

Since the stationary solvable model in fig. 5 is not center, entropy, or RMDP solvable, we can conclude it is not trivially or naively solvable.

### A.1.4 Not stationary solvable

Finally, we show that when we change $r_?$ to 70 and keep the bounds on $p \in [0.1, 0.9]$ the same for the stationary solvable model in fig. 5, we end up with a model (the *None of the above* model in fig. 5) that is neither stationary, nor naively, nor trivially solvable. We know that nature can guarantee a value of $r_? = 70$ by playing $p^{\circ} \in [0.3, 0.4]$ and $p^{\circ} \in [0.65, 0.7]$. We therefore know that any agent policy that can guarantee a value of 70 is optimal as well. In particular, we know that $\pi_s^*$ is optimal, since this policy always results in a value of $r_? = 70$.

We again identify the three naive nature policies:

- $\theta_{\text{Center}}$ assigns $p^{\circ} = p^{\circ} = 0.5$, as $\text{Centroid}(p) = 0.5$.
- $\theta_{\text{Ent}}$ assigns $p^{\circ} = p^{\circ} = 0.5$, as this creates the maximum entropy between states $s_7$ and $s_8$, namely $\{s_7 \mapsto 0.5, s_8 \mapsto 0.5\}$.
- $\theta_{\text{RMDP}}$ assigns $p^{\circ} = p^{\circ} = 0.9$, as in the fully observable model, it is always optimal for nature to minimize the chance of reaching state $s_8$, and therefore to maximize $p$.

We first construct the optimal agent policy $\pi'$ against the center and entropy nature policies, which are the same:

$$
\begin{aligned}
\pi' &= \operatorname*{argmax}_{\pi \in \Pi} V^{\pi,\theta_{\text{Center}}} \\
&= \operatorname*{argmax}_{\pi \in \Pi} \Big( r_? + (50 - 0.5r_?)a^{\circ} - 50a^{\circ}p^{\circ} - 25a^{\circ}a^{\circ\circ} + 75a^{\circ}p^{\circ}a^{\circ\circ} \\
&\qquad\qquad + (100 - 0.5r_?)a^{\circ} - 100a^{\circ}p^{\circ} - 100a^{\circ}a^{\circ\circ} + 150a^{\circ}p^{\circ}a^{\circ\circ} \Big) \\
&= \operatorname*{argmax}_{\pi \in \Pi} \Big( 70 + (50 - 0.5 \cdot 70)a^{\circ} - 50a^{\circ} \cdot 0.5 - 25a^{\circ}a^{\circ\circ} + 75a^{\circ} \cdot 0.5 \cdot a^{\circ\circ} \\
&\qquad\qquad + (100 - 0.5 \cdot 70)a^{\circ} - 100a^{\circ} \cdot 0.5 \cdot -100a^{\circ}a^{\circ\circ} + 150a^{\circ} \cdot 0.5 \cdot a^{\circ\circ} \Big) \\
&= \operatorname*{argmax}_{\pi \in \Pi} \Big( 70 + 15a^{\circ} - 25a^{\circ} - 25a^{\circ}a^{\circ\circ} + 37.5a^{\circ}a^{\circ\circ} \\
&\qquad\qquad + 65a^{\circ} - 50a^{\circ} - 100a^{\circ}a^{\circ\circ} + 75a^{\circ}a^{\circ\circ} \Big) \\
&= \operatorname*{argmax}_{\pi \in \Pi} \Big( 70 - 10a^{\circ} + 12.5a^{\circ}a^{\circ\circ} + 15a^{\circ} - 25a^{\circ}a^{\circ\circ} \Big)
\end{aligned}
$$

Since we are maximizing, it is optimal to set $a^{\circ}$, $a^{\circ}$, and $a^{\circ\circ}$ to 1 and $a^{\circ\circ}$ to 0. This $\pi'$ results in value $V^{\pi',\theta_{\text{Center}}} = 70 - 10 + 12.5 + 15 - 0 = 87.5$, whereas $\pi_s^*$ results in value $V^{\pi_s^*,\theta_{\text{Center}}} = 70$. However, $\pi'$ is not optimal against the entire set of nature policies $\Theta$, as nature can achieve a value of $< 70$

when playing a nature policy $\theta'$ with $4p^{\circ} - p^{\circ} > 2.2$:

$$
\begin{aligned}
V^{\pi',\theta'} &= 70 + 15a^{\circ} - 50a^{\circ}p^{\circ} - 25a^{\circ}a^{\circ\circ} + 75a^{\circ}p^{\circ}a^{\circ\circ} \\
&\quad + 65a^{\circ} - 100a^{\circ}p^{\circ} - 100a^{\circ}a^{\circ\circ} + 150a^{\circ}p^{\circ}a^{\circ\circ} \\
&= 70 + 15 \cdot 1 - 50 \cdot 1 \cdot p^{\circ} - 25 \cdot 1 \cdot 1 + 75 \cdot 1 \cdot p^{\circ} \cdot 1 \\
&\quad + 65 \cdot 1 - 100 \cdot 1 \cdot p^{\circ} - 100 \cdot 1 \cdot 0 + 150 \cdot 1 \cdot p^{\circ} \cdot 0 \\
&= 70 + 15 - 50p^{\circ} - 25 + 75p^{\circ} + 65 - 100p^{\circ} - 0 + 0 \\
&= 125 + 25p^{\circ} - 100p^{\circ} \\
&= 125 - 25(4p^{\circ} - p^{\circ}) \\
&< 125 - 25 \cdot 2.2 \\
&= 125 - 55 \\
&= 70
\end{aligned}
$$

The optimal agent policy $\pi'$ against the center and entropy policies cannot guarantee a value of 70 against the entire set of nature policies $\Theta$, hence we can conclude that $\pi'$ is not an optimal agent policy in the *None of the above* model in fig. 5 and that this model is not center or entropy solvable, nor trivially solvable.

We continue with the RMDP nature policy $\theta_{\text{RMDP}}$. We again construct the optimal agent policy $\pi'$:

$$
\begin{aligned}
\pi' &= \operatorname*{argmax}_{\pi \in \Pi} V^{\pi,\theta_{\text{RMDP}}} \\
&= \operatorname*{argmax}_{\pi \in \Pi} \Big( r_? + (50 - 0.5r_?)a^{\circ} - 50a^{\circ}p^{\circ} - 25a^{\circ}a^{\circ\circ} + 75a^{\circ}p^{\circ}a^{\circ\circ} \\
&\qquad\qquad + (100 - 0.5r_?)a^{\circ} - 100a^{\circ}p^{\circ} - 100a^{\circ}a^{\circ\circ} + 150a^{\circ}p^{\circ}a^{\circ\circ} \Big) \\
&= \operatorname*{argmax}_{\pi \in \Pi} \Big( 70 + (50 - 0.5 \cdot 70)a^{\circ} - 50a^{\circ} \cdot 0.9 - 25a^{\circ}a^{\circ\circ} + 75a^{\circ} \cdot 0.9 \cdot a^{\circ\circ} \\
&\qquad\qquad + (100 - 0.5 \cdot 70)a^{\circ} - 100a^{\circ} \cdot 0.9 \cdot - 100a^{\circ}a^{\circ\circ} + 150a^{\circ} \cdot 0.9 \cdot a^{\circ\circ} \Big) \\
&= \operatorname*{argmax}_{\pi \in \Pi} \Big( 70 + 15a^{\circ} - 45a^{\circ} - 25a^{\circ}a^{\circ\circ} + 67.5a^{\circ}a^{\circ\circ} \\
&\qquad\qquad + 65a^{\circ} - 90a^{\circ} - 100a^{\circ}a^{\circ\circ} + 135a^{\circ}a^{\circ\circ} \Big) \\
&= \operatorname*{argmax}_{\pi \in \Pi} \Big( 70 - 30a^{\circ} + 42.5a^{\circ}a^{\circ\circ} - 25a^{\circ} + 35a^{\circ}a^{\circ\circ} \Big)
\end{aligned}
$$

Since we are maximizing, it is optimal to set $a^{\circ}, a^{\circ}, a^{\circ\circ}$, and $a^{\circ\circ}$ all to 1. This $\pi'$ results in value $V^{\pi',\theta_{\text{Center}}} = 70 - 30 + 42.5 - 25 + 35 = 92.5$, whereas $\pi_s^*$ results in value $V^{\pi_s^*,\theta_{\text{Center}}} = 70$. However, $\pi'$ is not optimal against the entire set of nature policies $\Theta$, as nature can achieve a value of $< 70$ when playing a nature policy $\theta'$ with $p^{\circ} + 2p^{\circ} < 1.8$:

$$
\begin{aligned}
V^{\pi',\theta'} &= 70 + 15a^{\circ} - 50a^{\circ}p^{\circ} - 25a^{\circ}a^{\circ\circ} + 75a^{\circ}p^{\circ}a^{\circ\circ} \\
&\quad + 65a^{\circ} - 100a^{\circ}p^{\circ} - 100a^{\circ}a^{\circ\circ} + 150a^{\circ}p^{\circ}a^{\circ\circ} \\
&= 70 + 15 \cdot 1 - 50 \cdot 1 \cdot p^{\circ} - 25 \cdot 1 \cdot 1 + 75 \cdot 1 \cdot p^{\circ} \cdot 1 \\
&\quad + 65 \cdot 1 - 100 \cdot 1 \cdot p^{\circ} - 100 \cdot 1 \cdot 1 + 150 \cdot 1 \cdot p^{\circ} \cdot 1 \\
&= 70 + 15 - 50p^{\circ} - 25 + 75p^{\circ} + 65 - 100p^{\circ} - 100 + 150p^{\circ} \\
&= 25 + 25p^{\circ} + 50p^{\circ} \\
&= 25 + 25(p^{\circ} + 2p^{\circ}) \\
&< 25 + 25 \cdot 1.8 \\
&= 25 + 45 \\
&= 70
\end{aligned}
$$

The optimal agent policy $\pi'$ against the RMDP nature policies cannot guarantee a value of 70 against the entire set of nature policies $\Theta$, hence we can conclude that $\pi'$ is not an optimal agent policy in the *None of the above* model in fig. 5 and that this model is RMDP solvable.

Finally, we show the *None of the above* model in fig. 5 is not stationary solvable. We therefore construct an agent policy $\pi'$ that guarantees a value of 75 against the set of stationary nature policies, which is better than the value $\pi_s^*$ achieves, therefore we can conclude that $\pi_s^*$ is not optimal against the set of stationary nature policies.

Let $\pi'$ assign 1 to $a^{\circ}$, $a^{\circ}$, and $a^{\circ\circ}$, and assign 0.5 to $a^{\circ\circ}$. We get the following value:

$$
\begin{aligned}
\forall \theta \in \Theta^{\mathrm{Sta}} : V^{\pi',\theta} &= r_? + (50 - 0.5r_?)a^{\circ} - 50a^{\circ}p^{\circ} - 25a^{\circ}a^{\circ\circ} + 75a^{\circ}p^{\circ}a^{\circ\circ} \\
&\quad + (100 - 0.5r_?)a^{\circ} - 100a^{\circ}p^{\circ} - 100a^{\circ}a^{\circ\circ} + 150a^{\circ}p^{\circ}a^{\circ\circ} \\
&= 70 + (50 - 0.5 \cdot 70) \cdot 1 - 50 \cdot 1 \cdot p^{\circ} - 25 \cdot 1 \cdot 1 + 75 \cdot 1 \cdot p^{\circ} \cdot 1 \\
&\quad + (100 - 0.5 \cdot 70) \cdot 1 - 100 \cdot 1 \cdot p^{\circ} - 100 \cdot 1 \cdot 0.5 + 150 \cdot 1 \cdot p^{\circ} \cdot 0.5 \\
&= 70 + 15 - 50p^{\circ} - 25 + 75 \cdot p^{\circ} \\
&\quad + 65 - 100 \cdot p^{\circ} - 50 + 75p^{\circ} \\
&= 75 + 25p^{\circ} - 25p^{\circ} \\
&= 75
\end{aligned}
$$

Where the last step follows from the restriction to stationary nature policies. $\pi'$ hence guarantees a higher value than $\pi_s^*$ against the set of stationary nature policies, so $\pi_s^*$ is not optimal against the set of stationary nature policies.

Next we show that $\pi_s^*$ is the only agent policy that can guarantee a value of 70 against the entire set of nature policies. Let $\pi''$ be an agent policy with $a^{\circ}, a^{\circ} \in (0,1]$, then we get:

$$
\begin{aligned}
\forall \theta \in \Theta : V^{\pi'',\theta} &= r_? + (50 - 0.5r_?)a^{\circ} - 50a^{\circ}p^{\circ} - 25a^{\circ}a^{\circ\circ} + 75a^{\circ}p^{\circ}a^{\circ\circ} \\
&\quad + (100 - 0.5r_?)a^{\circ} - 100a^{\circ}p^{\circ} - 100a^{\circ}a^{\circ\circ} + 150a^{\circ}p^{\circ}a^{\circ\circ} \\
&= 70 + (50 - 0.5 \cdot 70)a^{\circ} - 50a^{\circ}p^{\circ} - 25a^{\circ}a^{\circ\circ} + 75a^{\circ}p^{\circ}a^{\circ\circ} \\
&\quad + (100 - 0.5 \cdot 70)a^{\circ} - 100a^{\circ}p^{\circ} - 100a^{\circ}a^{\circ\circ} + 150a^{\circ}p^{\circ}a^{\circ\circ} \\
&= 70 + 15a^{\circ} - 50a^{\circ}p^{\circ} - 25a^{\circ}a^{\circ\circ} + 75a^{\circ}p^{\circ}a^{\circ\circ} \\
&\quad + 65a^{\circ} - 100a^{\circ}p^{\circ} - 100a^{\circ}a^{\circ\circ} + 150a^{\circ}p^{\circ}a^{\circ\circ} \\
&= 70 + a^{\circ}(15 - 50p^{\circ} - 25a^{\circ\circ} + 75p^{\circ}a^{\circ\circ}) \\
&\quad + a^{\circ}(65 - 100p^{\circ} - 100a^{\circ\circ} + 150p^{\circ}a^{\circ\circ})
\end{aligned}
$$

Since $a^{\circ}, a^{\circ} > 0$, the agent can only guarantee a of 70 if at least $\forall \theta \in \Theta$:

$$
15 - 50p^{\circ} - 25a^{\circ\circ} + 75p^{\circ}a^{\circ\circ} \geq 0 \vee 65 - 100p^{\circ} - 100a^{\circ\circ} + 150p^{\circ}a^{\circ\circ} \geq 0
$$

However, when $p^{\circ} \in (0.3, 0.4)$ and $p^{\circ} \in (0.65, 0.7)$ this requirement does not hold, for example when $p^{\circ} = 0.35$ and $p^{\circ} = \frac{2}{3}$:

$$
\begin{aligned}
15 - 50p^{\circ} - 25a^{\circ\circ} + 75p^{\circ}a^{\circ\circ} &= 15 - 50 \cdot 0.35 - 25a^{\circ\circ} + 75 \cdot 0.35 \cdot a^{\circ\circ} \\
&= 15 - 17.5 - 25a^{\circ\circ} + 26.25 \cdot a^{\circ\circ} \\
&= -2.5 + 1.25a^{\circ\circ} \\
&< 0
\end{aligned}
$$

Where the last step follows from $a^{\circ\circ} \in [0,1]$. And similarly:

$$
\begin{aligned}
65 - 100p^{\circ} - 100a^{\circ\circ} + 150p^{\circ}a^{\circ\circ} &= 65 - 100 \cdot \frac{2}{3} - 100a^{\circ\circ} + 150 \cdot \frac{2}{3} \cdot a^{\circ\circ} \\
&= 65 - 66\frac{2}{3} - 100a^{\circ\circ} + 100 \cdot a^{\circ\circ} \\
&= -1\frac{2}{3} \\
&< 0
\end{aligned}
$$

Hence, an agent policy can only guarantee a value of 70 by playing $a^{\circ}, a^{\circ} = 0$

Since $\pi_s^*$ cannot be found by considering the subset of stationary nature policies, while it is the only optimal policy against the entire set of nature policies, we can conclude that the *None of the above* model in fig. 5 is not stationary solvable.

## A.2 Benchmarks

Next, we provide proofs for the theorems related to our proposed benchmarks.

### A.2.1 Echo machine

Let us first restate the theorem in the main text:

**Theorem 1.** ECHO, *with* $p = 0.01$, $\bar{p} = 0.99$, $\delta = 0.1$ *and* $\gamma = 0.95$, *is not in any of the solvability classes defined in Section 3.1.*

To prove the theorem in the main text, we first prove two helping lemmas:

**Lemma 2.** *If* $0.5 \in \mathcal{P}$, *then the optimal adversarial stationary policy* $\theta_S^*$ *picks* $p = 0.5$.

*Proof.* Let $\Theta_{\text{delayed}}$ denote the set of nature policies that is not stationary, but for which the choice of $p$ only applies with a delay of 1 timestep, which means each $\theta \in \Theta_{\text{delayed}}$ must pick $p$ for step $t$ based on history $h_{t-1} = (..., a_{t-2}, o_t)$ and previous state $s_{1-t}$. We show that the Nash-optimal policy in this class always picks 0.5, which means it is stationary. Moreover, since $\Theta_{\text{delayed}}$ contains all stationary policies, this proves this policy is also the Nash-optimal stationary policy.

To do so, we first notice that the choice of any policy $\theta \in \Theta_{\text{delayed}}$ only has an impact at histories where the agent is in state $x'$ but the agent has some belief over $x$ and $x'$, or similarly with $y'$ and $y$. Let $h_t$ be such a history, and assume $s_t = x'$. In that case, denote $\theta(h_t, x) = p$, $h_{a \in \{x,y\}} = (h_t, a, \perp)$ and $h_{a \in \{x,y\}, i \in \{x,y\}} = (h_t, a, \perp, g, o_i)$. Furthermore, let $V^\pi(h, s)$ denote the value of policy $\pi$ against $\theta$ given history $h$ and state $s$. In that case, the agent has the following value function for history $h_t$

$$V^\pi(h_t, x') = \frac{1}{2}\pi(x \mid h_t)\pi(r \mid h_x)\Big(V^\pi\big((h_x, r, \perp), n_x\big) + V^\pi\big((h_x, r, \perp), n_y\big)\Big)$$

$$+ \frac{1}{2}\pi(y \mid h_t)\pi(r \mid h_y)\Big(V^\pi\big((h_y, r, \perp), n_x\big) + V^\pi\big((h_y, r, \perp), x_y\big)\Big)$$

$$+ \pi(x \mid h_t)\pi(g \mid h_x)\Big[pV^\pi(h_{xx}, x') + (1-p)V^\pi(h_{xy}, y')\Big]$$

$$+ \pi(y \mid h_t)\pi(g \mid h_y)\Big[pV^\pi(h_{y,x}, x') + (1-p)V^\pi(h_{y,y}, y')\Big]$$

We simplify this formula in three ways. Firstly, we collect all values that do not directly depend on the choice of $p$ (i.e., the top two lines) into a constant $C^\pi$. Secondly, using symmetry of the model, we define $V(h_{x,x}, x') = V(h_{y,y}, y') := V_{\text{same}}$ and $V(h_{x,y}, y') = V(h_{y,x}, x') := V_{\text{diff}}$. Lastly, we denote denote $\pi(x \mid h_t) = \pi_x$, and since no new information is contained in $h_x$ or $h_y$ we may assume $\pi(g \mid h_x) = \pi(g \mid h_y) := \pi_g$. Then, our formula simplifies as follows:

$$V^\pi(h_t, x') = C^\pi + \pi_g\Big(\pi_x\big(pV_{\text{same}} + (1-p)V_{\text{diff}}\big) + (1-\pi_x)\big((1-p)V_{\text{same}} + pV_{\text{diff}}\big)\Big)$$

We compute the partial derivatives of this function (with constant factor $\pi_g$ remove for readability) as:

$$\frac{\partial V^\pi(h_t, x')}{\partial \pi_x} = pV_{\text{same}} + (1-p)V_{\text{diff}} - (1-p)V_{\text{same}} - pV_{\text{diff}}$$

$$= (1-2p)(V_{\text{diff}} - V_{\text{same}})$$

$$\frac{\partial V^\pi(h_t, x')}{\partial p} = \pi_x V_{\text{same}} - \pi_x V_{\text{diff}} - (1-\pi_x)(V_{\text{diff}} - V_{\text{same}})$$

$$= (1-2\pi_x)(V_{\text{diff}} - V_{\text{same}})$$

We find that there exists a saddle point at $\pi_x = p = 0.5$. Thus, for any history $h_t$ leading to $x'$, we find that $\theta_{\text{delayed}}^*(h_t, x) = 0.5$, and the same holds for $y'$ via symmetry of the model. Lastly, we note that the choice of $p$ is only relevant in these two states, which means the choice of $p = 0.5$ (or any other arbitrary choice) is optimal for other states. Thus, always picking $p = 0.5$ is Nash-optimal for our policy class $\Theta_{\text{delayed}}$, which proves our lemma. $\square$

**Lemma 3.** *For any history $h_t$, let $\tau_a, a \in A$ denote the highest $t$ at which the agent has picked action $a$. Then, the following history-based nature policy is optimal:*

$$\theta^*(h_t, s) = \begin{cases} \sup(\mathcal{P}) & \text{if } \tau_x > \tau_y \\ \inf(\mathcal{P}) & \text{otherwise.} \end{cases} \tag{4}$$

**Corollary 2.** *Let $h_t$ denote some history such that the $s_t \in \{x, y, x', y'\}$. Then, the optimal agent policy is as follows:*

$$\pi^*(h_t) = \begin{cases} x & \text{if } 1 - \sup(\mathcal{P}) > \inf(\mathcal{P}) \\ y & \text{otherwise} \end{cases}$$

*Proof.* The choice of $\theta$ only matters if $s = n'$, in which case either $\tau_x = t - 1$ or $\tau_y = t - 1$. We can denote the two value function for these cases as follows:

$$V^\pi(h_x, n') = \pi(r \mid h_x)C^\pi + \gamma\pi(g \mid h_x)\Big(pV^\pi(h_{x,x}, x') + (1-p)V^\pi(h_{x,y}, y')\Big)$$

$$V^\pi(h_y, n') = \pi(r \mid h_y)C^\pi + \gamma\pi(g \mid h_y)\Big(pV^\pi(h_{y,x}, x') + (1-p)V^\pi(h_{y,y}, y')\Big)$$

Since $x'$ and $y'$ have the same successor states and immediate rewards, the difference between $V^\pi(h_{x,x}, x')$ and $V^\pi(h_{x,y}, y')$ depends only on the history. We see that history $h_{x,y}$ (and $h_{y,x}$) give the agent strictly more information than $h_{x,x}$ (and $h_{y,y}$): for the former we know we are in $x'$ (and $y'$), while for the latter we could be in either $x$ or $x'$ (and $y$ or $y'$) with non-zero probability. Since more information can only allow the agent to pick better actions, we conclude $V^\pi(h_{x,x}, x') \leq V^\pi(h_{x,y}, y')$ (and $V^\pi(h_{y,y}, y') \leq V^\pi(h_{y,x}, x')$), and thus that the value is minimized for $p = 1 - \sup(\mathcal{P})$ for history $h_x$ (and for $p = \inf(\mathcal{P})$ for history $h_y$). The corollary holds via the same argument. $\square$

Next, we show that the optimal policy is suboptimal against all stationary policies. More precisely, we give conditions for which, against the worst-case nature policy, it is only optimal to repair if the agent has detected that the machine is broken, i.e., if it takes action $a_x$ but observes $y$ two timesteps later, or similarly for $s_y$ and $x$. In contrast, we give a similar condition such that, for the worst-case non-stationary policy, repair is optimal without detecting that the machine is broken. We formalize this logic as follows:

**Theorem 5.** *In the maintenance benchmark, let $\theta$ be any stationary nature policy, and $h_t$ be any history such that the agent has some some non-zero probability $< 1$ to be in state $n'$. Then, the optimal policy against $\theta$ chooses action $\pi(h_t) = g$ as long as:*

$$\frac{(1-\delta)}{1 - 0.5\gamma^2} + \frac{1 - \gamma^2\delta}{1 - \gamma^2(1 + \gamma\delta - \delta)}\left[\frac{0.5\gamma^2}{1 - 0.5\gamma^2} - 1\right] > 0. \tag{5}$$

*Proof.* Via our lemma above, we may assume $p = 0.5$, in which case our problem is a standard POMDP for which we can talk about beliefs. (Note that if $p \notin \mathcal{P}$, then the agent can always get strictly more information, and thus has less incentive to repair without seeing a malfunction.) We first note that if the agent has detected the machine is broken since it's last repair, then no history exists such that the probability of being in state $n'$ lies between 0 and 1. In that case, we may denote the two most 'extreme' beliefs possible as follows. $b_\top$ denotes any belief such that $b(n_x) + b(n_y) = 1$ (due to symmetry, these states are interchangeable for both nature and the agent), and $b_\perp$ denotes the belief such that $b(n')$ is maximized. In particular, let $n$ denote the number of number of times the agent has observed $x$ or $y$ since the last repair action, or the beginning of the episode if no repair action has yet occured. Then, we calculate the probability of being in state $n'$ as follows:

$$b_\perp(n') = \delta\sum_{n=1} 0.5^n \leq \frac{0.5\delta}{1 - 0.5} = \delta, \tag{6}$$

We remark that if repairing is optimal for any belief $h_t$, then it must be optimal in $b_\perp$. Thus, we consider the values for both actions $g$ and $r$ in $b_\perp$:

$$Q(b_\perp, g) := Q_g = (1 - \delta) + \gamma^2\big[0.5Q_g + 0.5Q_r\big]$$

$$= \frac{(1 - \delta) + 0.5\gamma^2 Q_r}{1 - 0.5\gamma^2}$$

$$Q(b_\perp, r) := Q_r = -1 + \gamma Q(b_\top, g)$$

To determine when measuring is optimal, we may look at the difference between these two values. In particular, action $g$ is optimal as long as the following holds:

$$Q_g - Q_r = \frac{(1 - \delta)}{1 - 0.5\gamma^2} + Q_r \left[ \frac{0.5\gamma^2}{1 - 0.5\gamma^2} - 1 \right] \geq 0 \tag{7}$$

$$\tag{8}$$

We note that $\frac{0.5\gamma^2}{1 - 0.5\gamma^2} < 1$ for all $\gamma$. Thus, we can use an upper bound for $Q_r$ to find an overapproximation of when measuring is optimal. For this, we use the expected value given the fully observable setting, which we denote as $V_{\text{RMDP}}$. We note that our theorem is trivially true if repairing is not worth it in the fully observable case, thus we may assume that $V_{\text{RMDP}}(n') = \gamma V_{\text{RMDP}}(n_x) - 1$. Then, we get:

$$V_{\text{RMDP}}(n_x) = V_{\text{RMDP}}(n_y) = 1 + \gamma^2 \Big( \delta V_{\text{RMDP}}(n') + (1 - \delta) V_{\text{RMDP}}(n_x) \Big)$$

$$= 1 + \gamma^2 \Big( \delta(\gamma V_{\text{RMDP}}(n_x) - 1) + (1 - \delta) V_{\text{RMDP}}(n_x) \Big)$$

$$= 1 - \gamma^2 \delta + \gamma^2 V_{\text{RMDP}}(n_x) \Big( 1 + \delta\gamma - \delta \Big)$$

$$= \frac{1 - \gamma^2 \delta}{1 - \gamma^2(1 + \gamma\delta - \delta)}$$

$$\geq Q_r$$

Filling this in for $Q_r$, we find that:

$$Q_g - Q_r \leq \frac{(1 - \delta)}{1 - 0.5\gamma^2} + \frac{1 - \gamma^2 \delta}{1 - \gamma^2(1 + \gamma\delta - \delta)} \left[ \frac{0.5\gamma^2}{1 - 0.5\gamma^2} - 1 \right]$$

Action $g$ is only optimal as long as this value is $\geq 0$, which gives us our bound. $\qquad \square$

**Theorem 6.** *In the maintenance benchmark, let $\theta^*$ be the optimal history-based nature policy, and define $q := 1 - \min \big( \inf(\mathcal{P}), 1 - \sup(\mathcal{P}) \big)$ Then, there exists a history $h_t$ such that the agent has a non-zero, but $< 1$, probability of being in state $n'$ but $\pi(h_t) = r$, as long as:*

$$\frac{1}{1 - \gamma(1 - \delta)} \left[ 1 - \frac{(1 - q)\gamma^2}{1 - q\gamma^2} \right] - \frac{(1 - \delta)}{1 - q\gamma^2} \geq 0 \tag{9}$$

*Proof.* Without loss of generality, assume $1 - \sup(\mathcal{P}) \geq \inf(\mathcal{P})$, in which case lemma 3 and it's corollarary state $\pi^*$ always picks action $x$ and $\theta^*$ chooses $p = \sup(\mathcal{P}) := q$. (The proof for $1 - \sup(\mathcal{P}) \leq \inf(\mathcal{P})$ follows via symmetry of the model.) In that case, following the same logic used in the proof of theorem 5, we define the belief with the highest probability to be in state $n'$ as

$$b_\perp(n') = \delta \sum_{n=1} q^n \leq \frac{q\delta}{1 - q}$$

We define $Q_g$ and $Q_r$ as before, which yields the following condition:

$$Q_r - Q_g = Q_r \left[ 1 - \frac{(1 - q)\gamma^2}{1 - q\gamma^2} \right] - \frac{(1 - \delta)}{1 - q\gamma^2} \geq 0$$

Since $\forall g, \gamma \in (0, 1)$: $\frac{(1-q)\gamma^2}{1-q\gamma^2} < 1$, we can use a lower bound on $Q_r$ to find a sufficient condition. One such lower bound is given by taking an agent policy that never measures, in which case

$$Q_r \geq \sum_{n=0} \big( \gamma(1 - \delta) \big)^{2n} = \frac{1}{1 - \gamma(1 - \delta)},$$

which yields the condition given in the theorem. $\qquad \square$

Lastly, the proof of theorem 1 follows from the fact that the parameters satisfy the conditions of both theorem 5 and theorem 6.

### A.2.2 Parity

We start by restating our theorem for convenience.

**Theorem 2.** PARITY$(\infty)$, with $P_1 = \{0.2\}$, $P_2 = [0.1, 0.7]$, $P_3 = [0.1, 0.7]$, and $\gamma \geq 0.7\bar{3}$, is not naively solvable.

*Proof.* We assume that the agent always picks an action according to its current most likely parity, which is clearly optimal. Thus, we can summarize the agent's uncertainty as the using the *even-odd ratio* $e = \max(\Pr(\text{even}), \Pr(\text{odd}))$. Since the agent never receives any information in PARITY, it follows that any optimal policy must be *cyclic*. More precisely, let $\pi_n$ denote a policy that repeatedly takes $n$ s-actions, then a normal action which resets $e$ to 1. Then, any optimal policy must be representable as $\pi_n$, for some $n \in \mathbb{N}$.

We show that the value of $\pi_0$ is higher than that of any other $\pi_n$. First, the value of $\pi_0$ is given as:

$$V^{\pi_0} = \sum_{n=0}^{\infty} \gamma^n = \frac{1}{1-\gamma} \tag{10}$$

If we can find any choice of probabilities for which any policy $\pi_{n \neq 0}$ has a lower value, then we are done. We start with $\pi_1$, for which we pick $p_1 = 0.2, p_2 = 0.5$ and $p_3 = 0.3$. In that case:

$$V^{\pi_1} \leq \left( p_1(1+\gamma) + p_2(2-2\gamma) + p_3(3+\gamma) \right) \sum_{n=0}^{\infty} \gamma^{2n}$$
$$= \frac{2.1 - 0.5\gamma}{1-\gamma^2} \tag{11}$$

This value is smaller than that of $\pi_0$ for $\gamma > \frac{11}{15} = 0.7\bar{3}$. For larger values of $n$, we assume that the adversary starts off with the choice above, then picks $p'_1 = 0.1, p'_2 = 0.1, p'_3 = 0.7$ as long at the agent keeps taking stochastic actions. We claim that under the second set of dynamics, $e$ converges to the value of $\frac{10}{19} \approx 0.526$. To prove this, we invoke Banach's fixed point theorem [48]. Denote $e'$ as the even-odd ratio after a single step under the dynamics above, then $e' = -0.9e + 1$. Using the $L^{\infty}$ distance, we find the following:

$$\left| e' - \frac{10}{19} \right| = \left| -0.9e + 1 - \frac{10}{19} \right| = 0.9 \left| e - \frac{10}{19} \right| \leq \left| e - \frac{10}{19} \right| \tag{12}$$

Thus, using Banachs theorem, $e$ converges to $\frac{10}{19}$. In particular, this means that the maximum $e$ that will be reached is given as $e_{\max} = 0.5 + \left| 0.5 - \frac{11}{19} \right| = \frac{11}{19} \approx 0.579$. Next, we note that the expected immediate return for the stochastic actions can be given as follows:

$$\mathbb{E}[r|e] = 2.1e - 2(1-e) = 4.1e - 2. \tag{13}$$

For our maximum value of $e$, this yields an immediate return of $\frac{71}{190} \approx 0.37$. Thus, for any $n > 1$, we write the value function as follows:

$$V^{\pi_n} \leq \left[ \sum_{t=0}^{\infty} \gamma^{nt+t} \cdot 2.1 + \left[ \sum_{t'=1}^{n-1} \gamma^{nt+t'} \frac{71}{190} \right] + \gamma^{(n+1)t}(3e-2) \right] < V^{\pi_0} \tag{14}$$

Thus, $\pi_0$ is optimal.

Next, we show that the expected value for $\pi_1$ is higher than that of $\pi_0$ for all naive nature policies, which implies that $\pi_0$ is not the policy with the highest value. Starting with $\theta_{\text{Center}}$ and $\theta_{\text{Ent}}$, we find that both pick parameters $p_1 = 0.2, p_2 = p_3 = 0.4$, in which case:

$$V^{\pi_1, \theta_{\text{Center}}} = V^{\pi_1, \theta_{\text{Ent}}} = \frac{2.2 - 0.2\gamma}{1-\gamma} \tag{15}$$

For $\theta_{\text{RMDP}}$, we find the parameters $p_1 = 0.2, p_2 = 0.7, p_3 = 0.1$. In contrast to the other policies, this means the most likely parity does not change after a stochastic step, and the expected value is given as:

$$V^{\pi_1, \theta_{\text{RMDP}}} = \frac{2.1 + 0.2\gamma}{1-\gamma} \tag{16}$$

Both values are strictly larger than $\frac{1}{1-\gamma}$ for any $\gamma \in [0, 1]$, which proves the model is not naively solvable. $\qquad \square$

## B   Evaluating Agent Policies in RPOMDPs

In this appendix, we provide proof for theorem 3, which we restate for convenience:

**Theorem 3.** *For any agent policy $\pi \in \Pi$ there exists a best-response nature policy $\theta_\pi^* \colon X \to \mathcal{U}$ such that $\forall x \in X \colon \mathcal{P}\big(\theta_\pi^*(x)\big) \in$ Extremes $\left(\{\mathcal{P}(u) \mid u \in \mathcal{U}\}\right)$.*

*Proof.* We first write out the value function for a policy $\pi$ against an optimal nature policy $\theta_\pi^*$:

$$
V^{\pi,\theta_\pi^*}(h_t, s, x) = \sum_{a \in A} \sigma(a \mid x) \inf_{u \in \mathcal{U}} \Bigg[ R(s,a)
$$
$$
+ \gamma \Big( \sum_{s' \in S} \sum_{o \in \Omega} \mathcal{P}(u)(s', o \mid s, a) \sum_{x' \in X} \tau(x' \mid x, o, a) V^{\pi,\theta_\pi^*}\big((h_t, a, o), s', x'\big) \Big) \Bigg]
$$

We notice that none of the terms in this formula depend on $h_t$ (except recursively via $V^{\pi,\theta}$), which means we can remove this dependency. With that, we rewrite the value function with simplified notation as follows:

$$
V^{\pi,\theta_\pi^*}(s, x) = \sum_{a \in A} \pi(a \mid x) Q^{\pi,\theta_\pi^*}\big(s, a, x, \theta_\pi^*(s, a, x)\big)
$$
$$
Q^{\pi,\theta_\pi^*}(s, a, x, u) = R(s,a) + \gamma \sum_{s' \in S} \sum_{o \in \Omega} \mathcal{P}(u)(s', o \mid s, a) \sum_{x' \in X} \tau(x' \mid x, o, a) V^{\pi,\theta_\pi^*}\big(s', x'\big)
$$
$$
\theta_\pi^*(s, a, x) = \arg \inf_{u \in \mathcal{U}} Q^{\pi,\theta_\pi^*}\big(s, a, x, u\big)
$$

Given this formula, we first show that an optimal nature policy exists with signature $\theta \colon X \to \mathcal{U}$. We start by defining the following nature policy:

$$
\theta_X(x) = \arg \inf_{u \in \mathcal{U}} \sum_{s \in S} \sum_{a \in A} Q^{\pi, \theta_\pi^*, X}\big(s, a, x, u\big) \tag{17}
$$

Recall our model assumes $(s, a)$-rectangularity, which implies $\mathcal{U}$ can be expressed such that every $(s, a)$ pair has a unique set of decision variables. Thus, there exists an $u \in \mathcal{U}$ that minimizes eq. (17) for each $(s, a)$ independently.

We prove that this choice is optimal, using proof by contradiction. If $\theta_X(x)$ is suboptimal for $x \in X$, then there must exist some history $h$ where its choice of $u$ is suboptimal. In particular, let $x$ be the first memory state reached where $\theta_X$ makes a suboptimal choice, i.e. where choosing decision variables according to $\theta_X$ and then following $\theta_\pi^*$ would lead to a higher value. In that case, there must be at least one state-action pair $s_{\text{diff}}, a_{\text{diff}} \in S \times A$ where the following holds:

$$
Q^{\pi,\theta_\pi^*}\big(s_{\text{diff}}, a_{\text{diff}}, x, \theta_\pi^*(s_{\text{diff}}, a_{\text{diff}}, x)\big) - Q^{\pi,\theta_X}\big(s_{\text{diff}}, a_{\text{diff}}, x, \theta_X(x)\big) < 0
$$

Since our model is $(s, a)$-rectangular, there exists a distinct set of decision variables that affect any state-action pair $\mathcal{P}(\cdot, \cdot \mid s, a)$, which we denote as $Var_{s,a}$. Then, denoting $p \in Var$ as a decision variable, we define the following memory-based nature policy:

$$
\theta_X'(x)(p) = \begin{cases} \theta_\pi^*(x)(p) & \text{if } p \in Var_{s_{\text{diff}}, a_{\text{diff}}} \\ \theta_X(x)(p) & \text{otherwise.} \end{cases}
$$

Looking at eq. (17), we see that $\theta_X'$ achieves the same same $Q$-values as $\theta_X$ for all state-action tuples $(s, a) \neq (s_{\text{diff}}, a_{\text{diff}})$, but achieves a lower value for $(s_{\text{diff}}, a_{\text{diff}})$. However, we had defined $\theta_X$ as the function that minimizes eq. (17), so we have a contradiction. This means no state-action tuple can exist where $\theta_X$ is suboptimal, in which case $\theta_X$ can never make a first suboptimal choice as compared to $\theta_\pi^*$ Thus, $\theta_X$ is optimal.

Next, we need only show that an optimal policy exists such that $\forall x \colon \theta_\pi^*(x) \in$ Extremes $(\mathcal{P})$. This immidiately follows from the definition of $Q^{\pi,\theta_\pi^*}$ with the observation that, if $\pi$ is fixed, $V^{\pi,\theta_\pi^*}$ is only dependent on the current choice $u$ via it's arguments $s'$ and $\tau(x, o, a)$. Thus, a variable assignment $u$ that greedily maximizes the probabilities of reaching tuples $(s', o)$ with low expected values is both optimal and complies with our condition. $\qquad\square$

# C   Efficient Approximations for Robust Agent Policies

Here, we provide extended analysis and proofs of the theoretical results in section 5. Let $\mathcal{B} \subset \Delta(S)$ be the finite set of reachable beliefs.

First, we introduce the $H_{\text{RPOMDP}}$ operator [45] in our notation. It will be used later in this appendix.

**Definition 5.** *$Q_{RPOMDP}$ is the fixed point of the operator $H_{RPOMDP}$, which is defined as:*

$$H_{RPOMDP}Q(b,a) = \sum_{s \in S} b(s) \left[ R(s,a) + \gamma \inf_{u \in \mathcal{U}} \sum_{o \in \Omega} \sum_{s' \in S} \mathcal{P}(u)(s', o \mid s, a) \max_{a' \in A} Q(\tau_u(b,a,o), a') \right], \tag{18}$$

*where $\tau_u$ is the belief update under variable assignment $u \in \mathcal{U}$ defined as:*

$$\tau_u(b,a,o)(s') = b'(s') \propto \sum_{s \in S} b(s)\mathcal{P}(u)(s', o \mid s, a) \tag{19}$$

Then, let us restate the definitions of the upper bounds we introduced in the main body of the paper.

**Definition 4.** *$Q_{RMDP}$ and $Q_{RFIB}$ are the fixed point of the operators $H_{RQMDP}$ and $H_{RFIB}$, defined as:*

$$H_{RQMDP}Q(b,a) = \sum_{s \in S} b(s) \left[ R(s,a) + \gamma \inf_{u \in \mathcal{U}} \sum_{s' \in S} \mathcal{T}(u)(s' \mid s, a) \max_{a' \in A} Q(\mathfrak{b}_{s'}, a') \right], and \tag{2}$$

$$H_{RFIB}Q(b,a) = \sum_{s \in S} b(s) \left[ R(s,a) + \gamma \inf_{u \in \mathcal{U}} \sum_{o \in \Omega} \max_{a' \in A} \sum_{s' \in S} \mathcal{P}(u)(s', o \mid s, a) Q(\mathfrak{b}_{s'}, a') \right]. \tag{3}$$

Below, we provide the reasoning for the uniqueness and existence of fixed points through the Banach fixed point theorem [48]. Then, it suffices to show that the operators are contraction mappings.

**Lemma 4.** *The fixed point $Q_{RMDP}$ is synonymous with the fixed point of robust dynamic programming on the fully observable RMDP and lifting the resulting values into belief space.*

**Corollary 3.** *Consequently, the operator $H_{RQMDP}$ is a contraction mapping, and the existence and uniqueness of the fixed point $Q_{RMDP}$ are guaranteed through the Banach fixed point theorem [23, 43].*

To prove that $H_{\text{RFIB}}$ is a contraction, we introduce the following two well-known lemmas.

**Lemma 5.** *Let $X$ be a compact set and $f, g$ be functions of type $X \to \mathbb{R}$. Then:*

$$\left| \sup_{x \in X} f(x) - \sup_{x \in X} g(x) \right| \leq \sup_{x \in X} |f(x) - g(x)|, and, \left| \inf_{x \in X} f(x) - \inf_{x \in X} g(x) \right| \leq \sup_{x \in X} |f(x) - g(x)|.$$

It is a well-established lemma that occurs relatively often. For a proof, see for instance [Lemma B.2; 30].

**Lemma 6** (Triangle Inequality)**.** *The* triangle inequality *states that, for any two real numbers $u, v \in \mathbb{R}$ the following inequality holds:*

$$|u + v| \leq |u| + |v|.$$

Now, we are set to state the main theorem to prove that $H_{\text{RFIB}}$ is indeed a contraction.

**Theorem 7.** *The operator $H_{RFIB} \colon (\mathcal{B} \times A \to \mathbb{R}) \to (\mathcal{B} \times A \to \mathbb{R})$ is a contraction mapping in terms of the infinity norm $\| \cdot \|_\infty$ and the discount factor $0 \leq \gamma < 1$ as Lipschitz constant.*

*Proof.* Let $b \in \mathcal{B}$ and $a \in A$ be any belief and action, and let $Q \colon \mathcal{B} \times A \to \mathbb{R}$ and $Q' \colon \mathcal{B} \times A \to \mathbb{R}$ be any two Q-functions. Then:

$$|H_{\mathrm{RFIB}}Q(b,a) - H_{\mathrm{RFIB}}Q'(b,a)| = \left| \sum_{s \in S} b(s) \Big[ R(s,a) + \gamma \inf_{u \in \mathcal{U}} \sum_{o \in \Omega} \max_{a' \in A} \sum_{s' \in S} \mathcal{P}(u)(s',o \mid s,a) Q(\mathfrak{b}_{s'},a') \Big] \right.$$

$$\left. - \sum_{s \in S} b(s) \Big[ R(s,a) + \gamma \inf_{u \in \mathcal{U}} \sum_{o \in \Omega} \max_{a' \in A} \sum_{s' \in S} \mathcal{P}(u)(s',o \mid s,a) Q'(\mathfrak{b}_{s'},a') \Big] \right|$$

$$\leq \gamma \sum_{s \in S} b(s) \left| \inf_{u \in \mathcal{U}} \sum_{o \in \Omega} \max_{a' \in A} \sum_{s' \in S} \mathcal{P}(u)(s',o \mid s,a) Q(\mathfrak{b}_{s'},a') \Big] \right.$$

$$\left. - \inf_{u \in \mathcal{U}} \sum_{o \in \Omega} \max_{a' \in A} \sum_{s' \in S} \mathcal{P}(u)(s',o \mid s,a) Q'(\mathfrak{b}_{s'},a') \right|$$

$$\leq \gamma \sum_{s \in S} b(s) \sup_{u \in \mathcal{U}} \left| \Big[ \sum_{o \in \Omega} \max_{a' \in A} \sum_{s' \in S} \mathcal{P}(u)(s',o \mid s,a) Q(\mathfrak{b}_{s'},a') \Big] \right.$$

$$\left. - \sum_{o \in \Omega} \max_{a' \in A} \sum_{s' \in S} \mathcal{P}(u)(s',o \mid s,a) Q'(\mathfrak{b}_{s'},a') \Big] \right|$$

$$\leq \gamma \sum_{s \in S} b(s) \sup_{u \in \mathcal{U}} \sum_{o \in \Omega} \left| \Big[ \max_{a' \in A} \sum_{s' \in S} \mathcal{P}(u)(s',o \mid s,a) Q(\mathfrak{b}_{s'},a') \Big] \right.$$

$$\left. - \max_{a' \in A} \sum_{s' \in S} \mathcal{P}(u)(s',o \mid s,a) Q'(\mathfrak{b}_{s'},a') \Big] \right|$$

$$\leq \gamma \sum_{s \in S} b(s) \sup_{u \in \mathcal{U}} \sum_{o \in \Omega} \max_{a' \in A} \left| \Big[ \sum_{s' \in S} \mathcal{P}(u)(s',o \mid s,a) Q(\mathfrak{b}_{s'},a') \Big] \right.$$

$$\left. - \sum_{s' \in S} \mathcal{P}(s',o \mid u,s,a) Q'(\mathfrak{b}_{s'},a') \Big] \right|$$

$$\leq \gamma \sum_{s \in S} b(s) \sup_{u \in \mathcal{U}} \left[ \sum_{o \in \Omega} \max_{a' \in A} \left[ \sum_{s' \in S} \mathcal{P}(u)(s',o \mid s,a) \Big| Q(\mathfrak{b}_{s'},a') - Q'(\mathfrak{b}_{s'},a') \Big| \right] \right]$$

$$\leq \gamma \sum_{s \in S} b(s) \sup_{u \in \mathcal{U}} \left[ \sum_{o \in \Omega} \max_{a' \in A} \left[ \sum_{s' \in S} \mathcal{P}(u)(s',o \mid s,a) \, ||Q - Q'||_\infty \right] \right]$$

$$= \gamma \, ||Q - Q'||_\infty$$

Thus, it follows that, making use of the definition of the infinity norm:

$$||H_{\mathrm{RFIB}}Q(b,a) - H_{\mathrm{RFIB}}Q'(b,a)||_\infty \leq \max_{\langle b,a \rangle \in \mathcal{B} \times A} |H_{\mathrm{RFIB}}Q(b,a) - H_{\mathrm{RFIB}}Q'(b,a)| \leq \gamma \, ||Q - Q'||_\infty.$$

$\square$

Lastly, we note that for $Q_{\mathrm{RFIB}}$ the set of reachable beliefs $\mathcal{B}_S$ considered by the operator is finite, as it contains only $|\mathcal{B}_S| = |S|$ unit beliefs. That is, the variable assignments $u$ chosen by nature do not lead to an explosion of the set of reachable beliefs. Therefore, computing the fixed point only requires computing iterations of the operator over $\mathcal{B}_S \times A$.

The following definition and theorem help establish the tightness of the heuristics. The proof largely follows that of Hauskrecht [21].

**Definition 6** (Monotone mapping). *A mapping $H \colon (\mathcal{B} \times A \to \mathbb{R}) \to (\mathcal{B} \times A \to \mathbb{R})$ is monotone if for any two $Q, Q'$ and for all $(b,a) \in \mathcal{B} \times A$, we have that $Q(b,a) \leq Q'(b,a) \to HQ(b,a) \leq HQ'(b,a)$.*

**Theorem 8** (Theorem 6, [21]). *Let $H_1$ and $H_2$ be two contractions mappings with fixed points $Q_1^*$ and $Q_2^*$, defined on the spaces of Q-value functions $\mathcal{Q}_1$ and $\mathcal{Q}_2$, respectively. We have $Q_2^* \geq Q_1^*$ if:*

- $Q_1^* \in \mathcal{Q}_2$ *and* $H_2 Q_1^* \geq H_1 Q_1^* = Q_1^*$,

- $H_2$ *is a monotone mapping.*

Note that we may have $\mathcal{Q}_1 \subset \mathcal{Q}_2$, i.e., $\mathcal{Q}_1$ may cover a smaller space of $Q$-value functions.

Now, we are set to prove the following theorem of the main paper:

**Theorem 4.** *Regarding tightness, the following inequalities on the fixed points hold:*

$$\forall b \in \Delta(S), \forall a \in A \colon Q_{RMDP}(b, a) \geq Q_{RFIB}(b, a) \geq Q_{RPOMDP}(b, a).$$

*Proof.* Let us first restate that it is known that $H_{\mathrm{RPOMDP}}$ is a contraction mapping [45]. Furthermore, note that it can be shown that the operators $H \in \{H_{\mathrm{RQMDP}}, H_{\mathrm{RFIB}}, H_{\mathrm{RPOMDP}}\}$ are monotone mappings, see for instance [Appendix B.1.4; 30]. Then, it follows from the following observation [21]:

$$
\begin{aligned}
H_{\mathrm{RQMDP}} Q(b, a) \quad &= \sum_{s \in S} b(s) \Big[ R(s, a) + \gamma \inf_{u \in \mathcal{U}} \sum_{s' \in S} \mathcal{T}(u)(s' \mid s, a) \max_{a' \in A} Q(\mathfrak{b}_{s'}, a') \Big] \\[2ex]
\geq H_{\mathrm{RFIB}} Q(b, a) \quad &= \sum_{s \in S} b(s) \Big[ R(s, a) + \gamma \inf_{u \in \mathcal{U}} \sum_{o \in \Omega} \max_{a' \in A} \sum_{s \in S} \mathcal{P}(u)(s', o \mid s, a) Q(\mathfrak{b}_{s'}, a') \Big] \\[2ex]
\geq H_{\mathrm{RPOMDP}} Q(b, a) \quad &= \sum_{s \in S} b(s) \Big[ R(s, a) + \gamma \inf_{u \in \mathcal{U}} \sum_{o \in \Omega} \sum_{s' \in S} \mathcal{P}(u)(s', o \mid s, a) \max_{a' \in A} Q(\tau_u(b, a, o), a') \Big].
\end{aligned}
$$

$\square$

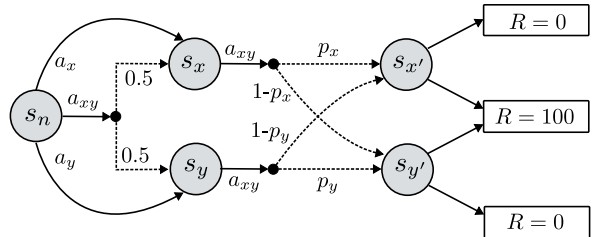

Figure 6: An example RPOMDP. We assume $p_x \in [0.7, 0.9]$ and $p_y \in [0.7, 0.9]$. We assume all states give the same observation.

## D  Experiments

### D.1  RHSVI

In this section, we give a brief overview of the RHSVI solver used in our experiments. We start with a brief introduction on value iteration for POMDPs, then repeat the initial formulation of RHSVI from Osogami [45] . Lastly, we describe several alterations made to RHSVI.

We provide an explanation of our alterations here for reproducibility. However, since we only use RHSVI as a baseline RPOMDP solver, we leave a complete description of the correctness and/or necessity of our alterations for future work.

**HSVI.** Since the value function for POMDPs is piecewise-linear [50], it can be represented as a (possibly infinite) set of linear lower bounds $\Gamma = \{\alpha \colon S \to \mathbb{R}\}$ known as $\alpha$-vectors. As shown [47], this set can be approximated by iteratively performing a *backup operation* on a finite set of beliefs $\mathcal{B}$. To find such a belief set, *heuristic search value iteration* [51, 52] builds a belief tree via sampling, guided by upper bounds on the value function. Thus, the belief set is restricted to reachable beliefs, which increases tractability. Moreover, by keeping track of an upper bound, the algorithm can determine when the policy it has found is $\epsilon$-optimal.

**Robust HSVI.** Osogami [45] generalizes HSVI to RPOMDPs by changing the robust backup operator and belief update function to robust variants. We repeat both here in this paper's notation. Given an uncertainty assignment $u$ and belief-action pair $(b, a)$, the backup operator is defined as follows:

$$\alpha_\Gamma(b, a, u)(s) = \sum_{o \in O} \alpha_{\Gamma, o}(b, a, u)(s) \tag{20}$$

$$\alpha_{\Gamma, o}(b, a, u) \in \operatorname*{argmax}_{\alpha \in \Gamma} \sum_{s, s' \in S} b(s) \mathcal{P}(u)(s', o \mid s, a) \alpha(s'). \tag{21}$$

Given this, we can define the robust backup function and the corresponding worst-case nature policy (which directly defines the belief update function) as follows:

$$\theta_\Gamma(b, a) \in \operatorname*{argmin}_{u \in \mathcal{U}} \sum_{s \in S} b(s) \alpha_\Gamma(b, a, u)(s), \tag{22}$$

$$\alpha_\Gamma(b, a) = \alpha_\Gamma(b, a, \theta_\Gamma(b, a)). \tag{23}$$

If our uncertainty set is given by intervals, then both can be computed using a linear program with at most $\mathcal{O}(|O||S|^2)$ variables and $\mathcal{O}(|S|^2|O| + |O||\Gamma|)$ constraints.[5] However, the support of both the belief and possible successor beliefs are often much smaller than $|S|$, which means the complexity mostly depends on $|O|$ and $|\Gamma|$ in practice.

Our implementation of RHSVI makes a number of changes from the description of [45], which we describe below:

**Robust backup.** First, we fix a problem with the backup procedure described above. Equation (21) implies that for each belief-action-observation tuple $(b, a, o)$, we can pick *any* alpha-vector that yields an optimal value against $\theta_\Gamma(b, a)$ to compute our backup. This yields alpha-vectors that give the

---

[5]For eq. (21), we implement the $\operatorname{argmax}$ operator via the constraint that $\sum_{s \in S} b(s) \alpha_\Gamma(b, a, \theta_\Gamma(b, a))(s)$ must be at least as high as any choice $\alpha \in \Gamma$.

correct value for the current belief, but may lead to problems if we use the alpha-vector for different beliefs.

To illustrate this problem, consider the RPOMDP shows in fig. 6. Working backwards, the following $\alpha$-vectors are can be found using the backup in states $s_{x'}$ and $s_{y'}$ and are thus valid:

$$\alpha_{x'}(s) = 100 \cdot [s = s_{x'}],$$
$$\alpha_{y'}(s) = 100 \cdot [s = s_{y'}].$$

Next, we use these alpha-vectors to perform backups in the beliefs $\mathfrak{b}_x$ and $b_{xy}$, which denote the beliefs reached from $s_n$ after action $a_x$ and $a_{xy}$, respectively. For the first, we can quickly see that the optimal nature policy picks $\{p_x = 0.7, p_y = 0.3\}$, which yields a value of 70. For belief $b_{xy}$, there are multiple optimal variable assingments for nature, including $\{p_x \mapsto 0.9, p_y \mapsto 0.9\}$. We use this assignment, in which case $\alpha_{x'}$ is a valid solution to eq. (21). In that case, our backup returns the following $\alpha$-vector:

$$\alpha_{xy}(s) = \begin{cases} 90 & \text{if } s = s_x \\ 10 & \text{if } s = s_y. \end{cases}$$

This gives us the correct value of $0.5$ for $b_{xy}$, but yields a value of $0.9$ for $\mathfrak{b}_x$, which is higher than the actual value of $0.7$ we computed before. Thus, even though $\alpha_{xy}$ can be found using the robust backup, it is not a valid underapproximation of the value function.

We discovered and empirically confirmed the problem with the backup function described above, but were unable to fully address the problem. Instead, we implement an ad-hoc fix that aims to ensure $\alpha_\Gamma(b, a)$ has the following properties:

1. *Indifference to state.* $\alpha_\Gamma(b, a)$ should have roughly equal values for each state in the support of the current belief.
2. *Indifference to nature.* $\alpha_\Gamma(b, a)$ should yield at least the same value against suboptimal nature policies, as compared to the optimal one.

To achieve both, we first perform the robust backup described above, which yields a nature policy $\theta^*$ and robust value $V$. We then solve a second LP to find an $\alpha$-vector $\alpha_r^*$ that yields the same value (up to a small error bound) but also has the properties above. To encode (1), we define the *state exploitability* of an $\alpha$-vector as follows:

**Definition 7.** *Given an $\alpha$-vector $\alpha$, belief $b$, and corresponding value $V_\alpha(b) = \sum_{s \in S} b(s)\alpha(s)$, the* state exploitability *of $\alpha$ with respect to $b$ is defined as:*

$$Expl(\alpha, b) = \sum_{s \in S} b(s)|\alpha(s) - V_\alpha(b)| \tag{24}$$

Intuitively, $Expl(\alpha, b)$ is zero if the expected value of $\alpha$ is independent of the actual state of the environment, while $Expl(\alpha, b) > Expl(\alpha', b)$ implies $\alpha$ is more robust against changes in the underlying state then $\alpha'$. We use $Expl(\alpha, b)$ as a *penalty term* for our LP, i.e., we aim to maximize $\left( \sum_{s \in S} b(s)\alpha_r^*(s) \right) - \delta e(\alpha_r^*, b)$ for some small value $\delta > 0$. To encode (2), we specify a number of nature policies that differ slightly from $\theta^*$, and add constraints so that $\alpha_r^*$ achieves at least value $V$ against all of these.

We empirically find that the approach above yields accurate value approximations, which is sufficient for the purposes of this paper. However, our testing has been limited to the experiments described in this paper, which were not specifically aimed at testing our backup. Apart from that, we have no theoretical basis to claim the approach is correct. We leave a more systematic analysis and solution to future work.

**Policy randomization.** Osogami [45] focuses on finding the value function for an RPOMDP, but does not consider the problem of constructing a policy which matches this value. In contrast to POMDPs, this is not a trivial problem, for similar reasons to the ones described above.

To illustrate this point, consider the environment of fig. 7, with $\delta \in [0, 1]$, and with $\gamma = 1$ for simplicity. Here, the agent's only meaningful action is to guess whether they are in state $x$ or $y$. Assuming $p \in [1, 0]$, $\theta_{\text{Nash}}$ should pick $p$ such that the value given for both actions $x$ and $y$ is equal. In this case, the agent is ambivalent about what action to pick, which means $\Gamma = \{\alpha_x\}$, with $\alpha_x(s) = [s = s_x]$, is a valid representation of the value function. However, a policy that always picks

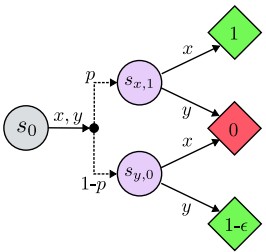

Figure 7: An RPOMDP showing the necessity of probabilities for robust policies.

action $x$ is clearly suboptimal, since it performs poorly against $\{p \mapsto 0\}$. Thus, it is not possible to create an optimal policy based on this $\Gamma$. From this example, we can conclude the following:

**Remark 2.** *To construct optimal RPOMDP policies, $\Gamma$ should generally include **all** $\alpha$-vectors corresponding to optimal actions for each reachable belief. In contrast, for POMDPs, **any** such $\alpha$-vector is sufficient.*

Note that we need not consider $\alpha$-vectors corresponding to *suboptimal* actions for reachable beliefs. With this condition satisfied, our policy should pick probabilities for each optimal action such that the expected value is robust against different nature policies. More precisely, we want to be robust against nature choosing different dynamics in previous timesteps, which would have led us to a different belief. Luckily, we have already defined the concept of state exploitability above, which aims to solve exactly this problem. Thus, we define our policy to aim and minimize state exploitability, which we formally define as follows:

$$d_\Gamma^{Expl} = \operatorname*{argmin}_{d \in \Delta(\Gamma)} Expl\Big(\big(\sum_{\alpha \in \Gamma^*} d(\alpha)\alpha\big), b\Big)$$

$$\alpha_\Gamma^{Expl}(s) = \sum_{\Gamma \in \Gamma^*} d_\Gamma^{Expl}\alpha(s)$$

$$\pi_\Gamma^{Expl}(a \mid b) = \sum_{\alpha \in \Gamma} d_\Gamma^{Expl}(\alpha)[\alpha \in \Gamma_a]$$

Although we have no proof that this is optimal, we show below that this choice is equivalent to minimizing a particular upper bound on the value function:

**Lemma 7.** *Let $\pi^*$ be an optimal policy and $\pi$ be a policy that picks actions with different probabilities for some belief $b$ but is otherwise identical. Then:*

$$V^{\pi^*} - V^\pi \le Expl(b, \pi). \tag{25}$$

*In particular, if $\forall b \colon Expl(b, \pi) = 0$, then $\pi$ is optimal.*

*Proof.* Assuming our model is graph-preserving, $\operatorname{supp}(b) = \operatorname{supp}(b')$. Let $s_+, s_-$ denote the states with the biggest difference in expected value, i.e.,

$$(s_+, s_-) \in \operatorname*{argmax}_{s, s' \in \operatorname{supp}(b)} \sum_{\alpha_a \in \Lambda^*(b)} \pi(a)\big[\alpha(s) - \alpha(s')\big]. \tag{26}$$

Recall that $\mathfrak{b}_s$ is the unit belief with $b(s) = 1$. Let $b_- = \mathfrak{b}_{s_-}$, and $b_+ = \mathfrak{b}_{s_+}$ in which case $V^\pi(b_-) \le V^\pi(b') \le V^\pi(b) = V^{\pi^*}(b) \le V^\pi(b_+)$. Thus, we rewrite eq. (25) as follows:

$$V^{\pi^*} - V^\pi \le V^\pi(b_+) - V^\pi(b_-)$$
$$= \sum_{\alpha_a \in \Lambda^*(b)} \pi(a)\big[\alpha(s_+) - \alpha(s_-)\big]$$
$$\le \operatorname{Exploit}(b, \pi),$$

which proves our lemma. □

**Computational optimizations.**

Next, we highlight a number of significant alterations to RHSVI that we make to improve performance:

*Belief Tree.* Equation (23) shows that the worst-case nature policy, and thus the belief update, depends on the current set of $\alpha$-vectors $\Gamma$. Thus, for any found belief $b$, the possible successor beliefs may change over time. This problem is not addressed by Osogami [45], which suggest they do not keep track of the belief tree at all. This is a valid approach, but it does not allow the reuse of belief nodes, which makes the method computationally expensive in practice. To deal with this, we use a belief tree, but periodically reset it at exponentially increasing intervals. This way, we guarantee that RHSVI finds all reachable beliefs (though this may take many iterations and resets in practice), while we can still use a tree structure to find new beliefs and compute tighter upper bounds more efficiently.

*Vector Pruning.* As in HSVI, we prune $\alpha$-vectors using point-wise domination with two changes. Firstly, since the complexity of both the backup- and belief update functions depend strongly on $|\Gamma|$, we try to keep this set as small as possible by only adding new $\alpha$-vectors if they are not dominated any $\alpha$-vectors in $\Gamma$ This requires additional overhead but drastically decreases the cost of backups. Secondly, to allow us to compute random policies, we only consider domination between $\alpha$-vectors that correspond to the same action. Thus, if $\alpha$ dominates $\alpha'$ but corresponds to a different action, then $\alpha'$ does not get pruned.

*Upper bounds.* We initialize the upper bounds with the robust upper bounds introduced in section 5 of the main body of the paper.

## D.2 Benchmarks & Infrastructure

**Infrastructure.** All experiments were conducted in Julia (version 1.11.5) on the same Ubuntu machine (version 22.04.5 LTS), which has an Intel(R) Core(TM) i9-10980XE CPU @ 3.00GHz and 256GB RAM (8 x 32GB DDR4-3200). We parallelize experiments across three workers [56].

**Benchmark descriptions.** Our new benchmarks, TOY* and ECHO, as well as the finite and infinite chain environments PARITY(10) and PARITY($\infty$), are described in section 3.

For the second set of benchmarks, we lift several (classic) POMDPs from the literature into RPOMDPs: TIGER [6] (also used in [22, 41]), MINIHALLWAY [33], and ALOHA [24], as well as an expanded variant of HEAVENORHELL [4] (also used in [45]).

- TIGER: The classic problem where an agent has to decide between two doors to open. One door has a tiger behind it, with a high negative reward associated, and one door has a prize with a high reward. The agent only knows the initial state distribution; the tiger is initialized to one of the doors with uniform probability. It therefore has to find a balance between noisy listening actions to gather information and deciding to open one of the doors. Therefore, it is intuitive that the nature policy heuristics that maximizes the entropy of the agent's belief leads to a good approximation of the worst-case.

- MINIHALLWAY: A small POMDP problem where the agent must navigate a corridor with noisy observations of the goal location, where a reward is located.

- ALOHA($N$): The agent must set the parameters of a communication protocol formulated as a POMDP, parameterized by $N$. A detailed description can be found in Jeon et al. [24].

- HEAVENORHELL($N$): An agent is positioned in the middle of a corridor of length $2 \times N$. At one end of the corridor, the agent can pass either through a door on its left or its right, one of which yields a reward (heaven) and one a penalty (hell). The location of the reward is observable on a sign at the other end of the corridor. Thus, the agent must learn to first visit this sign, then remember the location while traversing to the other side of the corridor.

We construct these RPOMDPs such that for any $\mathcal{T}(u)$, any transition probability is less than $0.5$ times higher or lower than the nominal POMDP, with no alterations to the observation function.

For the last set, we added partial observability to two benchmarks from the RMDP literature: HEALTHDETECTION [16] and REPLACEMENT [10].

- HEALTHDETECTION: The agent must schedule different methods of medical screening for colorectal cancer (or CRC) for a patient. The dynamics of the model are based on

| Dimensions | $|S|$ | $|\Omega|$ | $|A|$ |
|---|---|---|---|
| TOY$^*$ | 9 | 2 | 2 |
| ECHO | 8 | 2 | 2 |
| PARITY($\infty$) | 9 | 1 | 4 |
| PARITY (10) | 12 | 1 | 4 |
| TIGER | 2 | 2 | 3 |
| HEAVENORHELL(5) | 28 | 11 | 4 |
| HEAVENORHELL(10) | 48 | 21 | 4 |
| MINIHALLWAY | 13 | 9 | 3 |
| ALOHA(10) | 30 | 30 | 9 |
| REPLACEMENT | 7 | 7 | 4 |
| HEALTHDETECTION | 378 | 9 | 3 |

Table 1: Dimensions of the RPOMDP benchmarks used in the experimental evaluation.

| Algorithm | RHSVI($M_{\text{Center}}$) | | RHSVI($M_{\text{Ent}}$) | | RHSVI($M_{\text{RMDP}}$) | |
|---|---|---|---|---|---|---|
| Metric | min. | std. | min. | std. | min. | std. |
| TOY$^*$ | 37.48 | 35.76 | 37.48 | 0.01 | 32.49 | 17.88 |
| ECHO | 19.31 | 0.01 | 19.30 | 0.01 | 21.12 | 0.00 |
| PARITY($\infty$) | 9.25 | 0.74 | 9.07 | 1.05 | 13.65 | 0.00 |
| PARITY (10) | 59.92 | 0.10 | 62.40 | 0.04 | 55.19 | 0.07 |
| TIGER | 19.35 | 0.01 | 19.35 | 0.01 | 13.12 | 0.02 |
| HEAVENORHELL(5) | $-24.04$ | 0.01 | $-21.55$ | 0.01 | $-24.04$ | 0.00 |
| HEAVENORHELL(10) | $-37.35$ | 0.01 | $-36.71$ | 0.01 | $-37.36$ | 0.02 |
| MINIHALLWAY | 0.76 | 0.00 | 0.76 | 0.00 | 0.76 | 0.00 |
| ALOHA(10) | 62.41 | 0.15 | 59.70 | 0.11 | 56.49 | 0.10 |
| REPLACEMENT | $-47.64$ | 0.76 | $-46.69$ | 0.82 | $-46.58$ | 0.50 |
| HEALTHDETECTION | $-5718.42$ | 36.29 | $-5661.49$ | 67.73 | $-5554.34$ | 32.23 |

Table 2: Detailed statistics for the evaluation of the naive nature policies. We report the worst value (min.) out of 5 runs as computed on the nature MDP using MCTS and the standard deviation (std.) of the values found by the 5 MCTS runs.

real medical data. In the original paper, the authors only consider a number of existing screening protocols, which they combine with their model to construct robust Markov chains. However, the model can also be interpreted as an RPOMDP.

- REPLACEMENT: the agent must schedule costly repairs for a machine, which is represented by a chain environment. To transform this model into an RPOMDP, we assume the agent cannot observe the state of the machine unless they pay a measurement cost. Such *active measure* environments have been considered both for POMDPs [19, 27] and RPOMDPs [28].

We use discount factor $\gamma = 1$ for TOY$^*$, of $\gamma = 0.99$ for ECHO en HEAVENORHELL, and of $\gamma = 0.95$ for all other environments.

**Benchmark dimensions.** Table 1 shows the dimensions of the benchmarks used in the experimental evaluation.

### D.3 Error margins

In the plot in the main body of the paper, we normalize and plot the worst result among the 5 MCTS runs on the nature MDP. In tables 2 and 3, we provide the raw value (not normalized) of the worst evaluation out of the 5 runs, together with the standard deviation of the set of values found in the 5 MCTS runs, for all the algorithms tested.

| Algorithm | RHSVI | | RQMDP | | RFIB | |
| Metric | min. | std. | min. | std. | min. | std. |
|---|---|---|---|---|---|---|
| TOY* | 69.99 | 0.00 | 32.49 | 28.27 | 62.47 | 0.01 |
| ECHO | 31.10 | 0.00 | 25.45 | 0.01 | 25.44 | 0.01 |
| PARITY($\infty$) | 20.00 | 0.00 | 7.98 | 0.02 | 20.00 | 0.00 |
| PARITY (10) | 62.71 | 0.00 | 38.89 | 1.60 | 62.71 | 0.00 |
| TIGER | 19.36 | 0.01 | −20.04 | 7.33 | 14.49 | 1.27 |
| HEAVENORHELL(5) | −21.55 | 0.01 | −63.76 | 0.00 | −63.76 | 0.00 |
| HEAVENORHELL(10) | −36.71 | 0.01 | −63.76 | 0.00 | −63.76 | 0.00 |
| MINIHALLWAY | 0.76 | 0.00 | 0.25 | 0.00 | 0.25 | 0.00 |
| ALOHA(10) | 46.96 | 1.98 | 57.32 | 0.27 | 53.39 | 0.90 |
| REPLACEMENT | −46.16 | 0.15 | −51.96 | 0.37 | −70.87 | 2.02 |
| HEALTHDETECTION | −5596.72 | 46.17 | −5671.29 | 36.57 | −5660.60 | 64.77 |

Table 3: Detailed statistics for the evaluation of the new baselines and RHSVI. We report the worst value (min.) out of 5 runs as computed on the nature MDP using MCTS and the standard deviation (std.) of the values found by the 5 MCTS runs.

