# OpenReview forum: "On Evaluating Policies for Robust POMDPs"
_NeurIPS.cc/2025/Conference — NeurIPS 2025 poster_

### Official Review · Reviewer_rqdZ · 2025-06-24

**Clarity:** 3
**Significance:** 4
**Originality:** 3
**Rating:** 5
**Confidence:** 5

**Summary:**

This paper constructs infrastructure for RPOMDP research. In particular, the authors design benchmarks for testing RPOMDP algorithms, baselines to compare those algorithms against, and a method for computing the performance of arbitrary policies (which is non-trivial in *R*POMDPs, because one has to find the POMDP that is worst for the policy in question). In order to do the latter efficiently, they prove a result showing that the worst-case POMDP must be an extreme element of the convex set of POMDPs that compose the RPOMDP. They show why the benchmarks they propose are better, in the sense of being less easily solved by naive methods, than common existing benchmarks.

**Questions:**

You consider the setting where nature is adversarially picking environment parameters, and nature can change those parameters every timestep, after looking at the agent's past actions. How does that differ from the following setting: nature looks as the agent's whole policy, and then picks (permanent) environment parameters once?

**Ethical Concerns:**

["NO or VERY MINOR ethics concerns only"]

**Final Justification:**

Thank you to the authors for answering my questions and addressing my concerns. I still recommend acceptance.

**Limitations:**

Yes, but I would also propose adding a note that the benchmarks are fairly toy. You can certainly defend this, but it may be good to mention.

**Quality:**

4

**Strengths And Weaknesses:**

Strengths: This paper lays out it its aims clearly, those aims are compelling and important, and it accomplishes those aims. In my view, the prevailing feeling in RL that it's enough to study MDPs has hamstrung the field; studying RPOMDPs seems necessary for the field of RL to (safely) make contact with reality.

Weaknesses: There were a few points where I was confused by the mathematical presentation.

It seems that the RPOMDP field is immature enough that we cannot really complex environments as benchmarks. I suppose that weakens this paper, since everything is still fairly toy. However, I suppose we build to harder benchmark environments when we have better solvers, and we get better solvers by having better benchmark environments.

Confusions with the presentation:

> "decision variables v ∈ Vars,a only affect transitions and observations in the state-action pair ⟨s, a⟩"

I don't understand what this means.

> Lastly, we assume P is convex, which is a common assumption for tractability reasons

P is a function from U to probability distributions. Absent a total order over probability distributions, what does it mean to say P is convex? Do you mean the set {P(u) : u \in U} is convex? Likewise, is Extremes(P) really Extremes({P(u) : u \in U})?

> V n = minθ∈Θ maxπ∈Π V π,θ

Why do you ever define this quantity or talk about the Nash equilibrium? In the rest of the paper, as far as I can tell, the min over environments is always inside the max over policies.

> Intuitively, if a model is Θ-bar-solvable, then we can find an optimal agent policy against Θ by finding the optimal policy against only Θ-bar ⊆ Θ.

I don't follow. If it's theta bar solvable, then there exist policies are that are optimal both against theta and theta bar, but it doesn't mean that all policies that are optimal against theta bar are also optimal against theta. Do you mean "we can potentially find an optimal policy"?

> Intuitively, an RPOMDP with a stationary nature policy represents a POMDP.

Isn't the objective different? If you represent an RPOMDP with a stationary nature policy as a POMDP, then a solver should attempt to optimize by taking an expectation over the uncertain feature of the state that corresponds to the nature policy. In an RPOMDP, the agent is not taking an expectation over the uncertain feature of the state (the fixed nature policy), but rather minimizing it.

> Definition 3.

Should R_n(<s, x, a>) = *negative* R(s, a)?

> Def. 4

"b" is not defined (although I knew what it meant).

---

> ### Author Rebuttal · Authors · 2025-07-31
>
> We thank the reviewer for their feedback and comments.
> We will incorporate their comments in the final version of the paper.
> First, we answer the reviewer's question.
>
> > You consider the setting where nature is adversarially picking environment parameters, and nature can change those parameters every timestep, after looking at the agent's past actions. How does that differ from the following setting: nature looks as the agent's whole policy, and then picks (permanent) environment parameters once?
>
> If nature picks permanent environment parameters, this corresponds to picking a POMDP with the exact same states as the RPOMDP, i.e., static uncertainty.
> Our problem, where nature can change the parameters every timestep, i.e., dynamic uncertainty, gives nature more power than with static uncertainty, as has been shown in (Bovy et al. (2023)), and is therefore more robust.
> Even if nature can pick the environment parameters after observing the agent's entire policy, nature can achieve a lower reward by choosing different environment parameters at different histories.
>
>
>
> Next, we aim to clarify some of the comments made in their review.
>
> > I don't understand what $Var_{s,a}$ means.
>
> When considering $s,a$-rectangularity, each uncertainty variable can only influence the transition and/or observation probabilities in one state-action pair.
> We can therefore partition the set of uncertainty variables into smaller sets $Var_{s,a}$ that each only influence a single state-action pair $(s,a)$.
> The entire uncertainty set can then be written as $\mathcal{U} = \set{Var \to \mathbb{R}} = \bigtimes_{s,a}\set{Var_{s,a} \to \mathbb{R}}$.
>
> One can think of the transition and observation functions to be of type $\mathcal{T} \colon S \times A \to (\mathcal{U} \to \Delta(S))$ and $\mathcal{O}  \colon S \times A \to (\mathcal{U} \to \Delta(Z))$.
> We'll clarify the notation of $Var_{s,a}$ in the final version of the paper
>
>  > What does it mean to say $\mathcal{P}$ is convex? Do you mean the set $\set{\mathcal{P}(u) : u \in \mathcal{U}}$ is convex? Likewise, is $Extremes(\mathcal{P})$ really $Extremes(\set{\mathcal{P}(u) : u \in \mathcal{U}})$?
>
> Indeed, we mean the set $\set{\mathcal{P}(u) : u \in \mathcal{U}}$ is convex and $Extremes(\set{\mathcal{P}(u) : u \in \mathcal{U}})$.
> We thank the reviewer for bringing this to our attention.
> We will fix this in the final version of the paper.
>
> > Why do you ever define $V^n = \min_{\theta\in\Theta} \max_{\pi\in\Pi} V^{\pi,\theta}$ or talk about the Nash equilibrium?
>
> We discuss the Nash equilibrium because, ultimately, it is what an RPOMDP solver aims to find, as it is considered the standard robust objective.
> We use it to explain the solvability classes.
> We also need to explain the Nash equilibrium to discuss the difference in policy types between the Nash equilibrium's nature policy and the best-response nature policy.
> We indeed use $V^n$ only in our definition of the Nash equilibrium and can simplify this notation in the final version.
>
>  > Do you mean "we can potentially find an optimal policy"?
>
> We mean that we can use the set of optimal agent policies against $\overline{\Theta}$ to find an optimal policy against $\Theta$, but we still need to evaluate all the agent policies to find whether these policies are optimal against $\Theta$.
> We will clarify this.
>
>  > Isn't the objective different when an RPOMDP with a stationary nature policy represents a POMDP.
>
> If the stationary nature policy is known, the objective reduces to optimizing the POMDP, which shares the same states as the RPOMDP, by taking an expectation over the uncertain feature of the state corresponding to the nature policy.
> Our statement is intended as a contrast to non-stationary nature policies, where, in combination with the RPOMDP, the POMDP has a far larger, possibly infinite, state space.
> We never assume that the nature policy is known, and the objective for the agent always remains to maximize against the worst-case nature policy.
> We will revise our statement to clarify this point.
>
>  > Should $R_{n}(\langle s, x, a \rangle) = - R(s, a)$?
>
> No, the objective in the nature MDP is computing a nature policy that minimizes the discounted reward.
> Therefore, we use the same reward function as in the original RPOMDP.
> We describe that the best-response nature policy given an agent policy $\pi$ is the $\arg\min_{\theta\in\Theta^\pi_n} V_n^{\pi\theta}$ on line 258.
> We will make the objective in the nature MDP more explicit.
>
>  > "b" is not defined (although I knew what it meant).
>
> We define $b \in \Delta(S)$ on line 98, but we will clarify 'recall $b\in\Delta(S)$ is an agent belief' before Definition 4.
>
> ## References
> Bovy et al. (2023): Eline M. Bovy, Marnix Suilen, Sebastian Junges, Nils Jansen: Imprecise Probabilities Meet Partial Observability: Game Semantics for Robust POMDPs. IJCAI 2024: 6697-6706

---

> > ### Comment · Reviewer_rqdZ · 2025-08-01
> >
> > Thank you for your clarifications and explanations.
> >
> > > We discuss the Nash equilibrium because, ultimately, it is what an RPOMDP solver aims to find, as it is considered the standard robust objective.
> >
> > That doesn't seem to be the position of this paper! The paper seems to take the position, and I am inclined to agree, that an RPOMDP should solve for the Stackelberg equilibrium (with nature playing second), not the Nash equilibrium. That is, the best-response nature policy is what's relevant, not the Nash equilibrium nature policy. I think a quick comment explaining why the best response nature policy is different from a Nash equilibrium policy would suffice for this paper rather than presenting the Nash equilibrium mathematically.
> >
> > I think I'm probably belaboring a minor point. I think you could make the presentation leaner without it, but please feel free to ignore this comment.
> >
> > > If the stationary nature policy is known, the objective reduces to optimizing the POMDP, which shares the same states as the RPOMDP, by taking an expectation over the uncertain feature of the state corresponding to the nature policy.
> >
> > But it's not known, as you say two sentences later. So I don't follow when you say, "Our statement is intended as a contrast to non-stationary nature policies." You make a compelling distinction between a known, stationary nature policy and an unknown, non-stationary nature policy. But it seems like the more interesting point is considering the distinction between an unknown, stationary nature policy and an unknown, non-stationary nature policy.

---

> > > ### Author Response · Authors · 2025-08-05
> > >
> > > We thank the reviewer for their detailed response, and would like to further clarify some of our answers in the rebuttal.
> > >
> > > > The paper seems to take the position, and I am inclined to agree, that an RPOMDP should solve for the Stackelberg equilibrium (with nature playing second), not the Nash equilibrium.
> > > That is, the best-response nature policy is what's relevant, not the Nash equilibrium nature policy.
> > > I think a quick comment explaining why the best response nature policy is different from a Nash equilibrium policy would suffice for this paper rather than presenting the Nash equilibrium mathematically.
> > >
> > > We agree that, since we assume that nature can observe the agent's action and choose a variable assignment based on this information, a Stackelberg equilibrium is appropriate as well.
> > > We do note that this assumption is not general for Robust POMDPs, for example in the static setting, where a fixed but unknown transition function is assumed and nature cannot respond to the actions of the agent, or in the s-rectangular setting (Wiesemann et al. (2013)), where nature also no longer observes the agent action before choosing a variable assignment.
> > > We therefore feel that the discussion of the Nash equilibrium has added value to make the connection between existing work and our work more clear.
> > > However, we agree that mentioning the Stackelberg equilibria would improve the paper: we will add this to our introduction of the objective and the discussion of the best-response nature policy.
> > >
> > > > So I don't follow when you say, "Our statement is intended as a contrast to non-stationary nature policies."
> > > You make a compelling distinction between a known, stationary nature policy and an unknown, non-stationary nature policy.
> > > But it seems like the more interesting point is considering the distinction between an unknown, stationary nature policy and an unknown, non-stationary nature policy.
> > >
> > > We apologize for any confusion our response may have caused.
> > > Our remark *"Our statement is intended as a contrast to non-stationary nature policies."* and our original comment in the paper *"Intuitively, an RPOMDP with a stationary nature policy represents a POMDP."* both concern the POMDP that you get when applying a nature policy to the RPOMDP.
> > > We agree that the more interesting distinction is that between an unknown stationary nature policy and an unknown non-stationary nature policy.
> > > In the literature, this distinction is known as static versus dynamic uncertainty, as discussed in more detail in Bovy et al. (2024).
> > > We will revise our original comment in the paper.
> > > Furthermore, we will introduce static RPOMDPs, i.e., RPOMDPs with unknown stationary nature policies, in the paragraph spanning L102 to L109 and highlight the distinction between an unknown stationary nature policy and an unknown non-stationary nature policy.
> > >
> > > ### References
> > > Bovy et al. (2023): Eline M. Bovy, Marnix Suilen, Sebastian Junges, Nils Jansen: Imprecise Probabilities Meet Partial Observability: Game Semantics for Robust POMDPs. IJCAI 2024: 6697-6706
> > >
> > > Wiesemann et al. (2013): Wolfram Wiesemann, Daniel Kuhn, Berç Rustem: Robust Markov Decision Processes. Math. Oper. Res. 38(1): 153-183 (2013)

---

> > > > ### Comment · Reviewer_rqdZ · 2025-08-05
> > > >
> > > > Thank you for these updates.

---

### Official Review · Reviewer_sEqX · 2025-07-02

**Clarity:** 4
**Significance:** 3
**Originality:** 4
**Rating:** 5
**Confidence:** 3

**Summary:**

In this work, authors present a new evaluation pipeline for RPOMDPs that itself has been an understudied problem. As authors describe RPOMDPs are naturally described as a two-agent games where agent policy picks and action and then nature policy adversarially picks the dynamics. Resulting in worst possible dynamics for a given policy. Authors describe three solvability classes for nature policy: trivial, naive and stationary. They define that such nature policies to be too easy and describe benchmarks that are _not_ in those two solvability classes. Best response (BR) is found to be a good substitute worst-case nature policy. This results in a nature MDP and is solvable using off-the-shelf tools.

**Questions:**

-

**Ethical Concerns:**

["NO or VERY MINOR ethics concerns only"]

**Final Justification:**

I am satisfied with the authors rebuttal. I have also read other reviewers opinions and followed the discussion.

**Limitations:**

Yes

**Paper Formatting Concerns:**

-

**Quality:**

4

**Strengths And Weaknesses:**

Strengths:
- Understudied problem is put into a strong theoretical and practical footing.
- Idea of defining naive nature policies as too simple and showing that benchmark policies are not in the naive category gives theoretical grounding on the using benchmark policies instead of naive policies in comparisons.
- Connection from RPOMDP to MARL (via best response natural policy) is brilliant, even though obvious idea.
- Experiments also gave new information about the relative strengths of different environments and the usefulness of RFIB as an adequate baseline.

Weaknesses:
- How we can say that the only categories of naive solvability in Section 3.1 are these three? Can there be possible other possible policies for \theta. And if it is possible to have other policies that can be considered naive, then does it somehow interact with the Theorems in Section 3.2?

---

> ### Author Rebuttal · Authors · 2025-07-31
>
> We thank the reviewer for their comments and feedback.
> We will incorporate their comments in the final version of the paper.
> We answer their questions below.
>
> > How we can say that the only categories of naive solvability in Section 3.1 are these three? Can there be possible other possible policies for $\theta$?
>
> We propose that these three categories represent solvability classes that are useful and intuitive to identify.
> This is evidenced by the fact that in our experimental evaluation, nature policies in these naive solvability classes work well on many existing benchmarks.
> Yet, we do not claim that it is exhaustive.
> Additional categories of solvability can be defined as naive if so desired.
> We will clarify this in the paper.
>
> > And if it is possible to have other policies that can be considered naive, then does it somehow interact with the Theorems in Section 3.2?
>
> This depends on the type of naive policy that would be considered.
> In particular, Theorem 1 implies that Echo is not stationary solvable, which means it could only become naively solvable if you add a new 'naive' non-stationary agent policy.
> In contrast, Parity is constructed with the particular naive nature policies in mind, and thus is more likely to become solvable given different nature policies.

---

> > ### Comment · Reviewer_sEqX · 2025-08-03
> >
> > Thanks a lot for authors diligent rebuttal. I am quite satisfied with the answers.

---

### Official Review · Reviewer_bffU · 2025-07-03

**Clarity:** 3
**Significance:** 2
**Originality:** 3
**Rating:** 4
**Confidence:** 3

**Summary:**

The authors present a set of intuitions, evaluation methods, algorithms and results on the problem of evaluating policies of agents in Robust POMDPs. They propose new environments to use as benchmarks that satisfy some intuitive (and some formal) properties desirable for the evaluation of policies, they justify why it should be enough to evaluate the policies against nature best response (and not agains nature’s optimal policy) and provide two extensions of existing value iteration algorithms applied to RPOMDPs. Finally they run the algorithms in the proposed benchmarks and discuss the results and implications.

**Questions:**

1. The intuition behind solvability is not very obvious. While I see that it provides an argument for finding optimal policies in smaller subsets of environment parameters and ensuring global optimality, I do not really see how to use it to derive these policies, and I also do not follow how does it apply to the idea of evaluating RPOMDP policies against best response parameters. Can the authors expand on this, and justify the need for considering the given solvability definition?
2. Similar to the question above, the idea of naive solvability seems arbitrary (which the authors seem to agree as per their own assessment). Furthermore, the authors propose this taxonomy of solvability in terms of the nature policies, but provide results about solvability for specific nature policies that are not necessarily the best response. How does the concept of solvability link to the proposed evaluation method?
3. The authors touch slightly on the topic, but I would like to further understand the utility in practice of RPOMDPs. As in other minmax, zero sum game frameworks, obtained policies can be extremely conservative as some (very unlikely) particular nature parameter may force the optimal agent policy to act very suboptimally in other regions of the parameter space. As this work provides benchmarking and evaluation baselines for RPOMDPs, what problems do the authors consider it can be useful for?
4. Regarding the scalability of solving the nature MDP, the authors state that MCTS fails to provide valid nature MDP solutions in larger RPOMDPs. Is there a way of addressing this, or is it an artefact of the complexity of the resulting MDPs?
5. Is the fact that Tiger and HeavenOrHell are 'naively solvable' a problem for general evaluation of RPOMDPs?

**Ethical Concerns:**

["NO or VERY MINOR ethics concerns only"]

**Final Justification:**

I consider the author's reply satisfactory. I still have some leftover doubts over the general applicability of their defined taxonomy, but overall it is relevant work.

**Limitations:**

The authors address briefly the limitations of their work.

**Paper Formatting Concerns:**

No concerns

**Quality:**

3

**Strengths And Weaknesses:**

## Strengths
- The problem is well motivated, and the contributions are clear.
- The paper is novel and covers an existing gap in the RPOMDP literature.
- The paper is clear, easy to follow and introduces interesting ideas.

## Weaknesses
- Some decisions or results in the paper are not well justified.
- Some more discussion is needed to add nuance to the relevance of the contributions.
- The implications and scalability of the work are not really clear.

---

> ### Author Rebuttal · Authors · 2025-07-31
>
> We thank the reviewer for their feedback and comments.
> We will incorporate their comments in the final version of the paper.
> Below, we answer their questions.
>
> ## Summary
>
> #### Intuition of solvability and relation to best-response
> We do not view the solvability classes as methods to construct optimal nature policies.
> We define the solvability classes to classify challenging benchmarks, and propose a best-response evaluation method to enable robust policy evaluation given an (arbitrary) agent policy.
> These are distinct concepts of the paper, but in our experimental evaluation, we use both concepts together.
> Specifically, to show that (1) some commonly used benchmarks belong to naive solvability classes, and (2) our best-response evaluation is sound.
>
> #### Real-world applicability
>
> There exist many real-world problems described by RPOMDPs.
> We argue that we need a means for thorough evaluation of (the robustness of) policies for problems described by RPOMDPs.
> For instance, previous work evaluated their algorithms on benchmarks that we show by using our solvability categorization do not capture the complexities of RPOMDPs.
> Our paper addresses this gap in the literature.
>
> ## Questions
>
> > The intuition behind solvability is not very obvious. While I see that it provides an argument for finding optimal policies in smaller subsets of environment parameters and ensuring global optimality, I do not really see how to use it to derive these policies, and I also do not follow how does it apply to the idea of evaluating RPOMDP policies against best response parameters. Can the authors expand on this, and justify the need for considering the given solvability definition?
>
> We would like to clarify that the solvability classes are defined to classify RPOMDP benchmarks: we do not view them as methods to construct optimal nature policies.
>
> Aside from providing theoretical insights (Section 3.2), we can empirically identify whether an RPOMDP benchmark belongs to any of the solvability classes.
> In our evaluation (left-hand side of Figure 4), on each benchmark, we:
> - construct nature policies that are naive (e.g., based on ignoring partial observability or by taking the center of the uncertainty set),
> - compute an (optimal) agent policy against said nature policies (if a naive nature policy is given, the problem reduces to a POMDP for the agent), and
> - evaluate these agent policies against the best-response nature policies.
>
> For instance, if the optimal agent policy against the naive nature policy has the same best-response value as the optimal RPOMDP policy (which considers the full range of potential nature policies), then we have evidence that an RPOMDP benchmark is in that solvability class.
>
> > Similar to the question above, the idea of naive solvability seems arbitrary (which the authors seem to agree as per their own assessment).
>
> Indeed, we postulate that these are some of the solvability classes that are useful to identify.
> This is evidenced by the fact that in our experimental evaluation, nature policies in these solvability classes work well on many existing benchmarks.
> Yet, we do not claim that it is an exhaustive classification.
>
> > Furthermore, the authors propose this taxonomy of solvability in terms of the nature policies, but provide results about solvability for specific nature policies that are not necessarily the best response. How does the concept of solvability link to the proposed evaluation method?
>
> As we explain in more detail in the previous answer, the concept of solvability is not directly linked to the proposed evaluation method.
> We use both concepts in our evaluation pipeline to test whether an RPOMDP benchmark is challenging (i.e., does not belong to a naive solvability class).
>
> If an RPOMDP benchmark is not naively solvable, a method that reports a different optimal value than the actual best-response value, \emph{i.e.}, worst-case value given the computed agent policy, indicates that a method is not able to deal with more complex structures that may appear in RPOMDPs.
>
> > The authors touch slightly on the topic, but I would like to further understand the utility in practice of RPOMDPs. As in other minmax, zero sum game frameworks, obtained policies can be extremely conservative as some (very unlikely) particular nature parameter may force the optimal agent policy to act very suboptimally in other regions of the parameter space. As this work provides benchmarking and evaluation baselines for RPOMDPs, what problems do the authors consider it can be useful for?
>
> RPOMDPs are designed to provide robust and conservative policies, as they are inspired by the \emph{robust optimization} field.
> Such policies are, for instance, often a requirement for safety-critical settings where the model cannot be fully specified.
> Our Echo benchmark, inspired by predictive maintenance problems, resembles such a setting.
>
> > Regarding the scalability of solving the nature MDP, the authors state that MCTS fails to provide valid nature MDP solutions in larger RPOMDPs. Is there a way of addressing this, or is it an artefact of the complexity of the resulting MDPs?
>
> The scalability of our evaluation method is indeed an artefact of the Nature MDP that we construct.
> Note that prior to our work, it was unclear how to perform evaluation in RPOMDPs efficiently in general.
> Furthermore, we are not restricted to using MCTS, and our formulation allows for robust policy evaluation in RPOMDPs to directly benefit from advances in solving continuous-state MDPs.
>
>
> > Is the fact that Tiger and HeavenOrHell are 'naively solvable' a problem for general evaluation of RPOMDPs?
>
> Yes, if one only evaluates on naively solvable benchmarks, one can't be sure of how the algorithm performs on benchmarks that do not belong to these solvability classes.
> Note that Tiger and Heavenhell are challenging POMDP benchmarks, but, as we show, this fact does not make them challenging RPOMDP benchmarks.
> To advance the field and establish with confidence what constitutes a good RPOMDP solver, we must have challenging benchmarks for RPOMDPs.

---

> > ### Comment · Reviewer_bffU · 2025-08-03
> > **Rebuttal acknowledgement**
> >
> > I thank the authors for their responses and for addressing my questions.

---

### Official Review · Reviewer_BVbs · 2025-07-04

**Clarity:** 3
**Significance:** 2
**Originality:** 2
**Rating:** 3
**Confidence:** 4

**Summary:**

This paper addresses the evaluation of agent policies in Robust Partially Observable Markov Decision Processes (RPOMDPs), a framework for sequential decision-making under partial observability and model uncertainty. RPOMDPs are conceptualized as two-player games where an agent selects actions, and "nature" adversarially chooses the system dynamics to minimize the agent's value. The core challenge lies in evaluating an agent policy, which necessitates finding a worst-case adversarial nature policy – a task that is computationally demanding.

**Questions:**

1. Would help if the authors can provide inputs on scalability?
2. Can the authors provide inputs on generalisability of benchmarks beyond solvability, especially in real world problems?
3. Are there any real world problems where this work will be useful?
4. Can the provided approaches be extended beyond rectangular uncertainty sets?
5. Theorem 3 and its discussion mention that best response nature policy is simpler than optimal nature policy. Can authors provide more quantitative or concrete examples of how this simplicity is important in practice?

**Ethical Concerns:**

["NO or VERY MINOR ethics concerns only"]

**Final Justification:**

I was looking through the papers provided by the authors in the final response. I find the papers are either from control theory (where researchers do not exactly use POMDPs ) or theoretical papers, where the applicability is quite superficial.

Leaving aside the aspect of applications, as that may not be as crucial. I find the scalability and theoretical guarantees on equilibrium are problematic. I am still not in favour of accepting the paper.

**Limitations:**

Yes

**Quality:**

2

**Strengths And Weaknesses:**

Strengths

1. Interesting take on RPOMDP research, focussing primarily on evaluation of policies.
2. Considering different uncertainty sets, rectangular and convex.

Weaknesses

1. Impact of the approximate nature of Montecarlo approaches is not mentioned. Given that approximate policies are found with MCTS type approaches, aren't there gaps in evaluation?
2. It would seem that the current approach can have scalability issues especially with larger environments (e.g., ALOHA, HEALTHDETECTION)

---

> ### Author Rebuttal · Authors · 2025-07-31
>
> We thank the reviewer for their feedback and comments.
> We will incorporate their feedback in the final version of the paper.
> We first summarize our response, then we respond in detail to the weaknesses and questions stated in the review.
>
> ## Summary
>
> #### Scalability
>
> Robust policy evaluation in RPOMDP is an inherently challenging problem, as it requires searching over possible nature policies.
> Thanks to Theorem 3, we know that for the best-response (read: worst-case) value, we need not consider full histories for nature but only the memory of the agent.
> We use this result in our formulation into a continuous-state nature MDP, which is a more manageable formulation and directly benefits from advances in that area.
>
> We acknowledge that policy evaluation is still challenging due to the continuous and potentially high-dimensional state-space.
> Specifically, we address this in our paper in the following way:
>
> - we study scalability and the impact of the used approximation in our experiments (Q1),
> - we address it in our limitation section (L346), and
> - we highlight it as a potential direction for future work (L376).
>
> #### Real-world applicability
>
> There exist many real-world problems described by RPOMDPs.
> We argue that we need a means for thorough evaluation of (the robustness of) policies for problems described by RPOMDPs.
> For instance, previous work evaluated their algorithms on benchmarks that we show by using our solvability categorization do not capture the complexities of RPOMDPs.
> Our paper addresses this gap in the literature.
>
>
>
> ## Response to comments
>
> > Impact of the approximate nature of Montecarlo approaches is not mentioned. Given that approximate policies are found with MCTS type approaches, aren't there gaps in evaluation?
>
> We agree with the reviewer that our approximate evaluation method can lead to evaluation gaps.
> We discuss the necessity of an approximation directly after introducing the nature MDP (L263), and its impact is mentioned in our limitations paragraph (L346-347).
> Furthermore, we devote Q1 of the experimental evaluation to study the impact of the approximation (and the resulting evaluation).
> To briefly recap this research question, we indeed observe a few cases (e.g., Aloha, HealthDetection) with evaluation gaps.
> Nevertheless, for most benchmarks, we do not see such a gap, which suggests the MCTS approximation performs adequately.
> To alleviate the problem fully, we mention that future work could design solving methods that make use of the structure of the nature MDP (L376-377).
>
> We respond to the following comment and subsequent question in a single answer:
>
> > It would seem that the current approach can have scalability issues especially with larger environments (e.g., ALOHA, HEALTHDETECTION)
>
> and
>
> > Would help if the authors can provide inputs on scalability?
>
> As discussed in Section 4, our evaluation method reduces the problem to solving a continuous-state MDP, which originates from having to evaluate policies with arbitrary (possibly continuous) memory representations.
> Thus, the limitations to scalability are the same as those for solving high-dimensional continuous-state MDPs.
> Our evaluation method is not tied to MCTS, but allows the use of any continuous-state MDP solver.
> In particular, this means that the scalability of our evaluation method will improve as continuous MDP solvers get better.
>
> We point out that without Theorem 3, the policy evaluation problem would involve an exhaustive search in the space of possible nature policies; a problem much harder and less scalable than our formulation of solving a continuous-state MDP.
>
> ## Response to the remaining questions
>
> > Can the authors provide inputs on generalisability of benchmarks beyond solvability, especially in real world problems?
>
> The main goal of the proposed benchmarks is to explain and test solvability classes.
> In our paper, we identify a gap in the literature; the benchmarks used to test the solvers do not sufficiently represent the complexities of RPOMDPs.
> We argue that this solvability is relevant in the real world.
> For example, in security problems, it is often reasonable to assume an attacker can alter their plan based on the prior observations and actions of the defender.
> If we model this attacker as the adversary in an RPOMDP, then such memory-based policies could not be expressed by an RPOMDP that is stationary solvable.
> To this extent, our evaluation pipeline can be used to determine whether a solver can be robust against such adversaries.
>
> While our benchmarks may be abstract, we note that the Echo benchmark is inspired by predictive maintenance problems, while the use of chain environments (such as Parity) for benchmarking is relatively common in RL (Castronovo et al., 2016; Bellinger et. al., 2021).
>
> > Are there any real world problems where this work will be useful?
>
> There exist many real-world problems where agents need to deal with both model and state uncertainty.
> Such problems include any type of planning in dynamic environments (as caused by, e.g., weather or other agents) or unknown environments (such as in RL, or when using approximate models).
> Thus, we argue that we need a means for thorough evaluation of (the robustness of) policies for such problems.
> This work is a first step in that direction.
>
> > Can the provided approaches be extended beyond rectangular uncertainty sets?
>
> Our construction extends to $s$-rectangular uncertainty sets by changing the nature MDP to one where the dependency on the agent's action in the state is omitted.
>
> The tractability of non-rectangular uncertainty sets is still an open problem in the less complicated setting of RMDPs (L349-350); it is unclear how to efficiently address the fact that nature's choices are coupled across different states.
>
> Although it may be conservative, assuming $(s,a)$-rectangularity is the most common (L349), as it provides a lower bound even if the problem at hand does not show any form of rectangularity.
> Therefore, if the rectangularity is unknown, it can be argued that assuming $(s,a)$-rectangularity is the most robust way to approach the problem.
>
> At this point, to the best of our knowledge, no solvers for non-rectangular or $s$-rectangular RPOMDPs exist.
> Therefore, we focus on $(s,a)$-rectangularity uncertainty sets.
>
>
> > Theorem 3 and its discussion mention that best response nature policy is simpler than optimal nature policy. Can authors provide more quantitative or concrete examples of how this simplicity is important in practice?
>
> The simplicity of the best-response nature policies originates from the fact that we focus on finding the worst-case value (worst-case nature policy) given an agent policy.
> We motivate evaluation w.r.t. the best-response with Example 1; the (Nash)-optimal nature policy typically does not result in the worst-case value for a particular agent policy.
> As a consequence of focusing on the best-response nature policy, we need not consider all possible agent policies, as would be required for the (Nash-)optimal nature policy.
> Moreover, Theorem 3 states that this policy only needs to be based on the agent's memory, as opposed to the full history.
> As a concrete example, consider an agent policy represented by a finite state machine with only $N$ nodes, such as those found by Cubuktepe et al. (2021) or Galesloot et al. (2024).
> Then, we need only consider nature's responses across this finite set of nodes.
>
> ## References
> Bellinger et al (2021): Colin Bellinger, Rory Coles, Mark Crowley, Isaac Tamblyn: Active Measure Reinforcement Learning for Observation Cost Minimization. Canadian AI 2021.
>
> Castronovo et al. (2016): Michael Castronovo, Damien Ernst, Adrien Couëtoux, Raphael Fonteneau: Benchmarking for Bayesian Reinforcement Learning. PLoS ONE 11(6):e0157088
>
> Cubuktepe et al. (2021): Murat Cubuktepe, Nils Jansen, Sebastian Junges, Ahmadreza Marandi, Marnix Suilen, Ufuk Topcu: Robust Finite-State Controllers for Uncertain POMDPs. AAAI 2021: 11792-11800
>
> Galesloot et al. (2024): Maris F. L. Galesloot, Marnix Suilen, Thiago D. Simão, Steven Carr, Matthijs T. J. Spaan, Ufuk Topcu, Nils Jansen: Pessimistic Iterative Planning for Robust POMDPs. EWRL 2024

---

> > ### Comment · Reviewer_BVbs · 2025-08-07
> >
> > Thanks to the authors for providing detailed responses. Here are some quick summary responses:
> >
> > 1. Scalability seems to be an issue. Authors have mentioned that there are many real world problems, but I could not find any references. Can I please request for you to provide references to papers?
> >
> > 2. MCTS induces a gap and the answer seems to be that when solvers get better, this gap will close. Not sure if that answers my question. If it is a game theoretic solution and there is a gap on best response, wouldn't that have an impact?
> >
> > 3. The answer to the generalisability question is fair.
> >
> > 4.  Understand the part about non-existence of non-rectangular uncertainty sets. However, I feel this is extremely restrictive in terms of applicability.
> >
> > 5.  The answer to the question on theorem 3 makes sense. No concerns here.

---

> > > ### Author Response · Authors · 2025-08-08
> > >
> > > We would like to thank the reviewer for their follow-up questions. We provide answers below.
> > > > Scalability seems to be an issue. Authors have mentioned that there are many real world problems, but I could not find any references. Can I please request for you to provide references to papers?
> > >
> > > Certainly, some applications of RPOMDPs inspired by real-world problems include: epidemic control (Nakao et al, 2021), aircraft collision [7,12], and satellite navigation [7].
> > > Furthermore, we can list numerous applications of RMDPs and POMDPS that consider either state or model uncertainty, but neglect the other due to scalability concerns.
> > > Such applications show both types of uncertainty, which could be captured by RPOMDPs in a principled way.
> > >
> > > **Applications of RMDPs**
> > > * In reinforcement learning, RMDPs can be used to express model uncertainty (Suilen et al., 2022; Derman et al., 2020).
> > > These methods are restricted to fully observable settings, even though many applications for RL are partially observable in practice.
> > > * RMDPs can be used to abstract fully-observable dynamical systems, making them more tractable (Badings et al., 2023; Lavaei et al., 2023).
> > > Similarly, RPOMDPs could be used to abstract partially observable dynamical systems.
> > > * RMDPs have been used to compute conservative values for existing medical diagnosis protocols under model uncertainty [13].
> > > RPOMDPs provide a principled way to address the inherent partial observability of such problems.
> > >
> > > **Applications of POMDPs**
> > > * In the introduction, we mention a number of applications of POMDPs, including wild-life monitoring [10], robotics [27], and predictive maintenance [30,31].
> > > In these settings, POMDP models are often constructed using limited data, but the resulting model uncertainty is not taken into account.
> > > RPOMDPs provide a method of doing so.
> > > * Works such as Lee et al. (2023) and Lambrechts et al. (2024) consider reinforcement learning in a partially observable setting where the agent can learn the states it has visited after each episode, thus allowing for a model-based approach.
> > > Both works formalize their approach using POMDPs, while an RPOMDP would better represent the inherent model uncertainty.
> > >
> > > We will expand the section on the relevance of RPOMDPs in our introduction and include the references mentioned above.
> > > > MCTS induces a gap and the answer seems to be that when solvers get better, this gap will close. Not sure if that answers my question. If it is a game theoretic solution and there is a gap on best response, wouldn't that have an impact?
> > >
> > > We discuss the impact of the gap between the best-response policy and our approximation in Q1 of our experimental section, as well as in our previous answer.
> > > Perhaps we have some misunderstanding.
> > > Could the reviewer clarify what we are missing or what is unclear in this discussion?
> > > > Understand the part about non-existence of non-rectangular uncertainty sets. However, I feel this is extremely restrictive in terms of applicability.
> > >
> > > We agree with the reviewer that the assumption of $(s,a)$-rectangularity can be restrictive.
> > > Nevertheless, many real-world problems can be expressed using $(s,a)$-rectangularity, including the examples of RMDP applications mentioned above.
> > > Furthermore, we reiterate that the assumption of $(s,a)$-rectangularity provides a valid lower value bound for any non-rectangular problem.
> > > Thus, $(s,a)$-rectangular problems are sufficiently expressive to warrant further investigation.
> > >
> > > We will add this argument to the paper (at L108).
> > >
> > > **References:**
> > >
> > > Badings et al. (2023): Thom Badings, Licio Romao, Alessandro Abate, David Parker, Hasan Poonawala, Marielle Stoelinga, Nils Jansen: Robust control for dynamical systems with non-Gaussian noise via formal abstractions. J. Artif. Intell. Res. 76, 341–391 (2023)
> > >
> > > Derman et al. (2020): Esther Derman, Danierl Makowitz, Timothy Mann, Shie Mannor: A Bayesian Approach to Robust Reinforcement Learning. UAI, 2020.
> > >
> > > Lee et al. (2023): Jonathan Lee, Alekh Agarwal, Christoph Dann, Tong Zhang: Learning in POMDPs is Sample-Efficient with Hindsight Observability. ICML, 2023.
> > >
> > > Lambrechts et al. (2024): Gaspard Lambrechts, Adrien Bolland, Damien Ernst: Informed POMDP: Leveraging Additional Information in Model-Based RL. RLC, 2024.
> > >
> > > Lavaei et al. (2023): Abolfazl Lavaei, Sadegh Soudjani, Emilio Frazzoli, Majid Zamani: Constructing MDP Abstractions Using Data With Formal Guarantees. IEEE Control Systems Letters, vol. 7, pp. 460-465, 2023
> > >
> > > Nakao et al. (2021): Hideaki Nakao, Ruiwei Jiang, Siqian Shen: Distributionally Robust Partially Observable Markov Decision Process with Moment-based Ambiguity. SIAM, 2021.
> > >
> > > Suilen et al. (2022): Marnix Suilen, Thiago D. Simão, David Parker, Nils Jansen: Robust Anytime Learning of Markov Decision Processes. NeurIPS, 2022.
> > >
> > > Song et al. (2022): Jun Song, William Yang, Chaoyue Zhao: Decision-dependent distributionally robust Markov decision process method in dynamic epidemic control. IISE Transactions Vol. 56, 2024, Issue 4.

---

> ### Author Response · Authors · 2025-08-07
>
> We would like to again thank the reviewer for their feedback and comments. Since the deadline for the discussion is approaching, we would like to ask whether our rebuttal has adequately addressed the questions and concerns raised in this review.

---

### Note · Authors · 2025-08-12

As a final remark, we would like to thank the reviewers for the fruitful discussion.
We were happy to hear that reviewers are mostly positive about the novelty, relevance, and clarity of the paper.
We will incorporate the feedback from the reviews, which should improve the clarity even further.
Next, we highlight the three most substantial changes:
* Reviewers `BVbs` and `bffU` had questions about the practical applicability of RPOMDPs.
In response, we will add a paragraph to our introduction including the references we mentioned in the discussion.
* Reviewers `BVbs` and `bffU` asked for clarification about the scalability of the proposed evaluation method, and reviewer `BVbs` asked for clarification about the approximation errors.
In response, we will clarify in the introduction and Section 4 that both the scalability and approximation error are a direct consequence of (approximately) solving a continuous MDP, which we believe is unavoidable.
* Reviewer `rqdZ` asked for more information about RPOMDPs with stationary nature policies.
These are known as *static RPOMDPs* in literature: we will add references to these models in Section 2.

We hope that our rebuttals, in combination with this final response, have been satisfactory.

---

### Decision · Program_Chairs · 2025-09-17

**Decision:**

Accept (poster)

**Comment:**

This paper studies an underexplored problem of evaluating policies for robust partially observable Markov decision processes (RPOMDP).  The paper introduces and formalizes the concept of solvability of RPOMDP, noting that finding the optimal policy for an RPOMDP is relatively easy when one only needs to consider a small set of nature policies.  The paper then proposes benchmarks based on the solvability of RPOMDPs.  The paper also proposes a way to evaluate an RPOMDP policy by reducing it to finding the optimal policy for continuous MDP.

Overall, the reviewers highly value the contributions, giving with two Accept and two Borderline ratings.  AC has also read the manuscript and confirmed the reviewers' evaluations.

The most notable weakness is insufficiency of the discussion on the approximate nature of the approach of solving the continuous MDP.  Overall, however, the strengths outweigh the weaknesses, and the AC confidently recommends an acceptance of the paper.